# Constitutive loss of DNMT3A causes morbid obesity through misregulation of adipogenesis

Ayala Tovy[1,2], Jaime M Reyes[1,2,3], Linda Zhang[1,2,4], Yung-Hsin Huang[1], Carina Rosas[1,2], Alexes C Daquinag[5], Anna Guzman[1,2], Raghav Ramabadran[1,2], Chun-Wei Chen[1,2,3], Tianpeng Gu[1,2], Sinjini Gupta[1,2], Laura Ortinau[3,6], Dongsu Park[3,6], Aaron R Cox[2,7], Rachel E Rau[1,8], Sean M Hartig[2,7], Mikhail G Kolonin[4], Margaret A Goodell[1,2]*

[1]Stem Cells and Regenerative Medicine Center, Baylor College of Medicine, Houston, United States; [2]Department of Molecular and Cellular Biology, Baylor College of Medicine, Houston, United States; [3]Department of Molecular and Human Genetics, Baylor College of Medicine, Houston, United States; [4]Graduate Program in Translational Biology and Molecular Medicine, Baylor College of Medicine, Houston, United States; [5]Institute of Molecular Medicine, McGovern Medical School at the University of Texas Health Science Center, Houston, United States; [6]Center for Metabolic and Degenerative Disease, Institute of Molecular Medicine, McGovern Medical School, University of Texas Health Science Center at Houston, Houston, United States; [7]Division of Endocrinology, Diabetes, and Metabolism, Department of Medicine, Baylor College of Medicine, Houston, United States; [8]Department of Pediatrics, Baylor College of Medicine and Texas Children's Hospital, Houston, United States

*For correspondence:
goodell@bcm.edu

Competing interest: The authors declare that no competing interests exist.

**Abstract** DNA Methyltransferase 3 A (DNMT3A) is an important facilitator of differentiation of both embryonic and hematopoietic stem cells. Heterozygous germline mutations in *DNMT3A* lead to Tatton-Brown-Rahman Syndrome (TBRS), characterized by obesity and excessive height. While DNMT3A is known to impact feeding behavior via the hypothalamus, here we investigated a role in adipocyte progenitors utilizing heterozygous knockout mice that recapitulate cardinal TBRS phenotypes. These mice become morbidly obese due to adipocyte enlargement and tissue expansion. Adipose tissue in these mice exhibited defects in preadipocyte maturation and precocious activation of inflammatory gene networks, including interleukin-6 signaling. Adipocyte progenitor cell lines lacking DNMT3A exhibited aberrant differentiation. Furthermore, mice in which *Dnmt3a* was specifically ablated in adipocyte progenitors showed enlarged fat depots and increased progenitor numbers, partly recapitulating the TBRS obesity phenotypes. Loss of DNMT3A led to constitutive DNA hypomethylation, such that the DNA methylation landscape of young adipocyte progenitors resemble that of older wild-type mice. Together, our results demonstrate that DNMT3A coordinates both the central and local control of energy storage required to maintain normal weight and prevent inflammatory obesity.

## Editor's evaluation

In this manuscript, the authors show that DNMT3A is essential for the regulation of adipocyte lipolysis and organismal metabolism. Focusing on a mouse model with heterozygous null mutations in DNMT3A, as seen in Tatton-Brown-Rahman syndrome in humans, the authors show that animals

develop obesity with adipocyte hypertrophy. Using a combination of single cell analysis, additional genetic models, and bisulfite sequencing, the authors propose a mechanism whereby DNMT3A regulates adipocyte differentiation and overall metabolism. This well-written paper will be of broad interest to researchers studying metabolism, development, and epigenetics.

## Introduction

De novo DNA methyltransferase 3 A (*DNMT3A*) plays a central role in establishing DNA methylation patterns during mammalian early embryogenesis and development (*Okano et al., 1999*). Deletion of *Dnmt3a* in mice results in postnatal lethality, confirming its necessity during development (*Li et al., 1992*). In humans, somatic mutations in *DNMT3A* are enriched in the hematopoietic system during aging and malignancy (*Ley et al., 2010*; *Tovy et al., 2020*). Loss of *DNMT3A* in hematopoietic stem and progenitor cells alters their epigenetic landscape, increasing the stem cell pool through self-renewal at the expense of differentiation (*Challen et al., 2011*). Whether DNMT3A regulates differentiation in other stem cell populations is largely unknown.

Germline heterozygous mutations in *DNMT3A* lead to Tatton-Brown-Rahman syndrome (TBRS; OMIM: 615879) (*Tatton-Brown et al., 2014*), a dominant disorder characterized by excessive height (~80% of individuals), intellectual disability (~80% of individuals), and obesity (~70% of individuals) (*Tatton-Brown et al., 2014*; *Tatton-Brown et al., 2018*). We and others have recently shown that *Dnmt3a* haploinsufficient mice can be used to study TBRS (*Christian et al., 2020*; *Tovy et al., 2022*). These mice recapitulate key TBRS phenotypes, including enlarged body size and obesity (*Christian et al., 2020*). However, the mechanisms through which DNMT3A regulates weight gain in TBRS are incompletely understood.

Obesity is a complex energy balance disorder reflecting deficient communication between the hypothalamus and adipose tissue, and only a few genes have so far been identified in which mutations exert causal roles in humans (*Choquet and Meyre, 2010*). Of these genes, the majority play a role in the central nervous system, which is pivotal in regulating behaviors controlling nutrient update and energy expenditure. Yet, although it is well known that mechanisms that impact adipocyte progenitor expansion and differentiation will also lead to obesity, the data on a potential role for DNA methylation in regulating adipose tissue are conflicting (*Ma and Kang, 2019*). While broad inhibition of DNA methyltransferases with compounds such as 5-aza-2'-deoxycytidine was reported to promote lipid accumulation in preadipocyte cell lines (*Yang et al., 2016*), DNMT inhibition in preadipocyte and mesenchymal precursor cell lines reduced adipogenic differentiation capacity (*Chen et al., 2016b*). Similarly, the role of *DNMT3A* in body weight regulation is unclear. While one study showed homozygous ablation of *Dnmt3a* specifically in hypothalamic Sim1-positive neurons led to hyperphagia and obesity (*Kohno et al., 2014*), knockout of *Dnmt3a* in mature adipocytes protected mice from developing insulin resistance while on a high-fat-diet (*You et al., 2017*).

Here, we utilize a mouse model of TBRS to establish that while dysregulation of *Dnmt3a* indeed regulates food intake, *Dnmt3a* also plays a cell-intrinsic role in adipocyte progenitors to permit normal maturation. We find that, similar to its role in the hematopoietic system, DNMT3A regulates adipose stem and progenitor cell function. Our results establish that DNMT3A directly regulates obesity through the adipose tissue, in addition to its role in the hypothalamus, and demonstrate how constitutive loss of *DNMT3A* creates a domino effect in which excess feeding together with faulty adipocytes lead to obesity.

## Results

### Heterozygous *Dnmt3a* loss leads to adipose tissue expansion and weight gain

TBRS patients harbor heterozygous frameshift or missense mutations in *DNMT3A*, predicted to be partial or complete loss-of-function, distributed throughout the gene's main functional domains (*Tatton-Brown et al., 2018*; *Tovy et al., 2022*). In order to study the role of DNMT3A in weight gain and adipose tissue biology, we utilized mice heterozygous for a *Dnmt3a* null allele ('3A-HET') along with their wild-type (WT) counterparts. In humans, TBRS phenotypes become particularly

evident around adolescence when patients exhibit marked height and weight gain (*Tatton-Brown et al., 2018*). Similarly, WT and 3A-HET mice were indistinguishable until around 3 months of age. However, by 6 months, both male and female 3A-HET mice were heavier than controls (*Figure 1A* and *Figure 1—figure supplement 1A*). At 12 months of age, the cohort of HET averaged 3 standard deviations (SD) above the mean weight of WT mice, becoming obese (*Figure 1B, C*). The increased length of 3A-HET mice, while statistically significant, was mild, and therefore could not account for the increased weight (*Figure 1—figure supplement 1B*). Body composition analysis was performed to determine whether fat versus lean mass contributed to the weight difference. We detected a steady increase in fat percentage and a concomitant reduction in lean mass in 3A-HET mice (*Figure 1C, D*, *Figure 1—figure supplement 1G*).

Adipose tissue can expand through either increased number of adipocyte cells due to proliferation and differentiation of adipocyte precursors (hyperplasia) or increased adipocyte cell size (hypertrophy). White adipose tissue (WAT) functions primarily to store energy, while brown adipose tissue (BAT) performs thermogenic functions. From WT and 3A-HET mice at 1 year of age, we dissected fat depots (gonadal, and subcutaneous inguinal white adipose tissue and brown adipose tissue) and performed histological analysis to determine size of BAT and WAT adipocytes. On average, 3A-HET mice had heavier fat depots across all measured depots (*Figure 1—figure supplement 1H*) and larger adipocytes in both WAT and BAT depots compared to WT littermates (*Figure 1E, F*), suggesting the increase in total fat is largely due to hypertrophy. Expansion of adipocytes is often accompanied by infiltration of macrophages (CD45 +, F4\80 + cells) and indeed we measured more F4/80 cells in WAT from HET mice compared to WT mice at 1 year of age (*Figure 1—figure supplement 1E*). These results suggest that reduced *DNMT3A* leads to pathological adipose expansion and spontaneous obesity.

Fatty liver is strongly associated with obesity, we therefore performed histological analysis of WT and 3A-HET mice livers at 1 year of age. We observe increase in fat accumulation in 3A-HET livers, as measured by fat-soluble dye (Oil-Red-O staining). Therefore, these results demonstrate that germline loss of DNMT3A leads to weight gain through expansion of adipose tissue, as well as accumulation of lipids in non-adipose tissues such as the liver (*Figure 1—figure supplement 1G, H*).

Together these data from heterozygous *Dnmt3a* KO mice mirror the findings in TBRS patients of increased size and marked obesity that develops after adolescence. Thus, we further examined the mechanisms accounting for obesity associated with reduced DNMT3A.

## *Dnmt3a*-deficient mice are hyperphagic and develop glucose and insulin resistance

We next tested whether 3A-HET mice showed increased food intake or decreased energy expenditure, or both, using a comprehensive lab animal monitoring system (CLAMS). We profiled WT and HET males and females at 6 months of age (*Figure 2—figure supplement 1A*). We measured increased food intake in HET mice mostly during the dark phase (*Figure 2A*). HET mice also showed a mild change in activity during light phase and decrease in energy balance calculated by calorie consumption compared to calorie expenditure (*Figure 2B* and *Figure 2—figure supplement 1C, D*). Other metabolic parameters were not altered (*Figure 2—figure supplement 1E–N*). As leptin secreted from adipocytes normally signals satiety to the hypothalamus to restrict food intake and inhibit weight gain (*Badman and Flier, 2007*; *Rosen and Spiegelman, 2014*), we measured leptin levels in 3A-HET compared to WT mice. We observed a persistent increase in leptin levels (*Figure 2—figure supplement 1G*) in mice lacking DNMT3A compared to their WT littermates. These data demonstrate that heterozygous germline loss of *Dnmt3a* impacts feeding behavior similarly to homozygous loss of *Dnmt3a* specifically in the hypothalamus (*Kohno et al., 2014*).

Prolonged weight gain can result in insulin resistance, which manifests as hyperinsulinemia combined with hyperglycemia and dyslipidemia. Thus, we tested whether 3A-HET mice displayed insulin resistance before (2 months) and after (6 months) weight gain. At 2 months, WT and HET mice did not display major differences in fasting glucose and insulin levels or when challenged with a glucose tolerance test (*Figure 2C–E*). However, at 6 months, when 3A-HET mice are statistically heavier than WT, they had higher fasting levels of glucose and insulin (*Figure 2E*). Further, at 10 months HET mice develop profound glucose and insulin resistance (*Figure 2F*). These results establish that loss of a

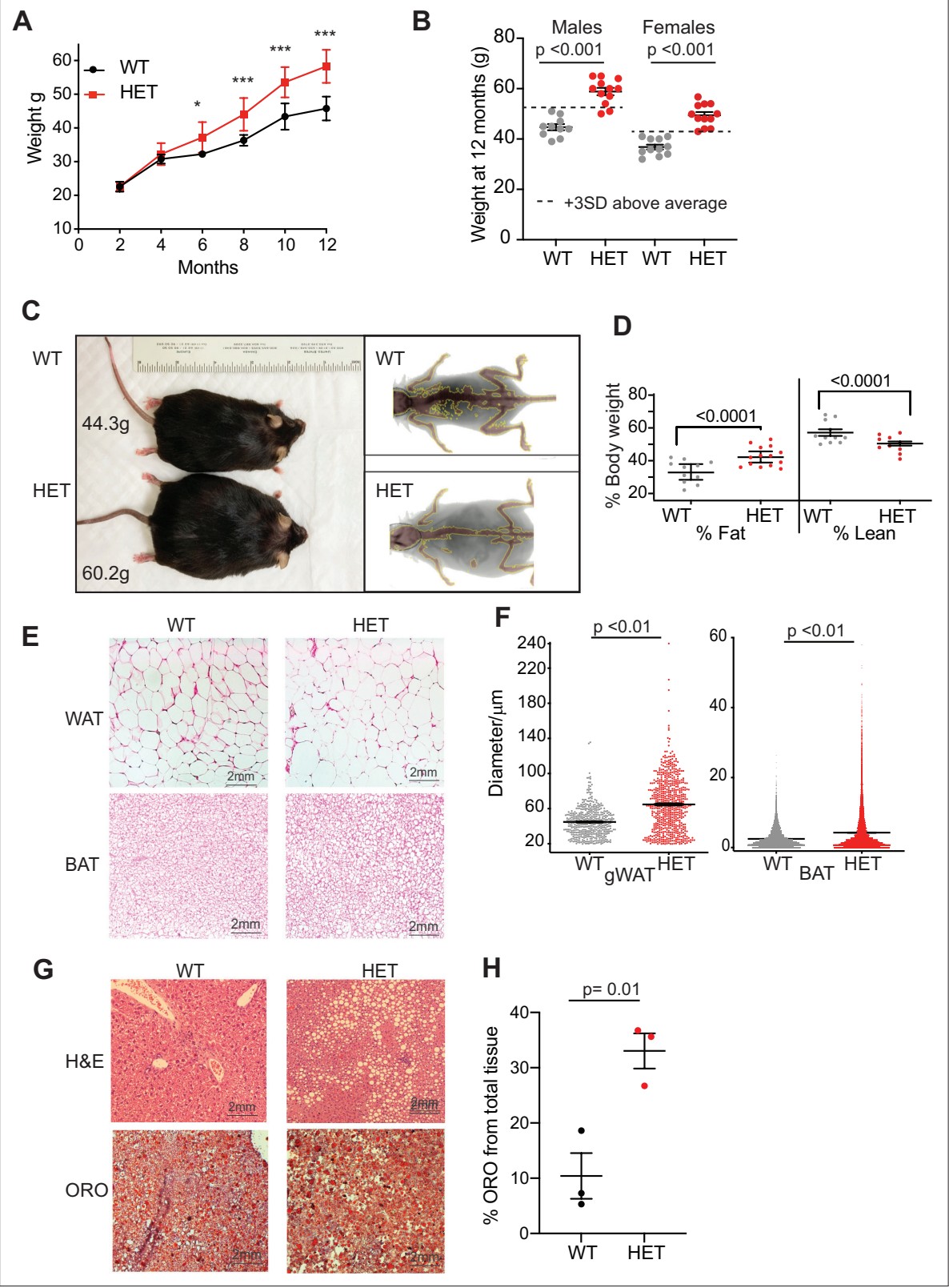

**Figure 1.** Dnmt3a-HET mice become overweight and develop obesity. (**A**) Weight of male *Dnmt3a*-HET (HET, n=14) and WT (n=12) littermates on regular chow. * p=0.03, ***<p = 0.001 as determined by two-way ANOVA. (**B**) Distribution of weight from HET and WT mice females and males at 1 year of age. Dashed lines indicate 3 standard deviations (SD) above the mean of the WT. (**C**) Representative display of D3A-HET and WT mice at 1 year Left: D3A-HET and WT mice at 1 year. Right: dual energy X-ray absorptiometry (DEXA) analysis of 6-month-old WT and HET mice. Top: Representative

*Figure 1 continued on next page*

*Figure 1 continued*

image displaying lean (brown color) and fat (grey) tissues. (D) Quantification of fat and lean mass (n=10–14) from DEXA. Statistical analysis 1way ANOVA. (E) Representative H&E staining of gonadal white adipose tissue and BAT from 1-year-old mice. (F) Morphometric analysis of gWAT and BAT adipocyte diameter. ***p<0.001 by 1way ANOVA. (G) Representative staining of liver tissue from 1-year-old WT and 3A-HET littermates. Top: H&E staining. Bottom: Oil Red O lysochrome diazo dye. (H) Quantification of total red droplets from Oil Red O-stained livers (n=3 per group). Statistical analysis one-way ANOVA.

The online version of this article includes the following source data and figure supplement(s) for figure 1:

**Source data 1.** List of cluster marker genes.

**Figure supplement 1.** *Dnmt3a*-HET mice recapitulate TBRS phenotypes.

---

*Dnmt3a* in this mouse model of TBRS leads to obesity and contributes to insulin resistance at older ages.

## DNMT3A promotes adipocyte differentiation and its loss increases progenitor pool

Given the marked obesity of *Dnmt3a* heterozygous mice, we considered the possibility that factors other than hypothalamus-regulated behaviors contribute additively to the increased adiposity. Because loss of DNMT3A can inhibit somatic stem cell differentiation (*Challen et al., 2011*), we scrutinized adipocyte maturation in 3A-HET mice.

Adipose tissue depots are composed of multiple cell populations including preadipocyte stem and progenitor cells, preadipocytes, and fully mature adipocytes, which change during development (*Merrick et al., 2019*).

To examine how loss of *Dnmt3a* affected these cell populations, we performed single-cell RNA sequencing (scRNAseq) on cells isolated from inguinal white adipose tissue (iWAT) of WT and 3A-HET mice at 8 weeks and 1 year old. We focused on the subcutaneous inguinal adipocytes as this depot was previously investigated by scRNAseq enabling the use of published data as reference for cluster annotation (*Merrick et al., 2019*). In addition, we selected these time points to capture adipose tissue cell states that span an expansion period (8 weeks) and obesity in 3A-HET mice (1 year), reflecting changes in adipose tissue and weight gain over time. We focused on WAT since its associated with increased risk for insulin resistance as observed in 3A-HET mice.

We first defined profiles in WT adipose tissue focusing on preadipocyte stem and progenitors. We separated stromal vascular cells (SVCs) from adipose tissue and then used flow cytometry to sort adipocyte stem and progenitor cells (*Figure 3A* and *Figure 3—figure supplement 1A*) and subjected those cells to scRNAseq. We performed standardized normalization and principal component analysis (PCA) on all conditions and from both genotypes (*Figure 3—figure supplement 1B, C*). To define component cell types, we next performed unsupervised clustering of gene expression of all cells and obtained six clusters (*Figure 3B*). We defined cluster marker genes as those differentially expressed in each cluster compared to all other clusters (*Figure 1—source data 1*, *Figure 3C, D*) and used those marker genes to define cluster identity using MSigDB pathway enrichment. Cluster 1 expressed key genes associated with committed preadipocytes, adipocyte differentiation, and inflammatory activation such as fatty acid binding protein (*Fabp4*), lipoprotein lipase (*Lpl*), and intercellular adhesion molecule–1 (*Icam1*) (*Balamurugan and Sterneck, 2013*). Cluster 2 displayed markers of committed differentiating preadipocytes (*Fabp4, Lpl, and Plin2*), as well as enrichment for the tissue factor *F3*, a marker of adipogenesis-regulatory cells associated with adipocyte differentiation (*Schwalie et al., 2018*).

We observed that both clusters 3 and 4 expressed a mix of stem and progenitor markers (*Dpp4, Cd34*) and adipocyte markers (*Fabp4, Lpl*), indicating they contained cells in transition. Importantly, dipeptidyl peptidase (*Dpp4*) is a marker of multipotent progenitors within iWAT with some attributes of mesenchymal stem cells (*Merrick et al., 2019*). Accordingly, pathway enrichment analysis of clusters 3 and 4 identified hallmarks of 'epithelial-to-mesenchymal transition' and 'adipogenesis' (*Figure 3C, D*, *Figure 3—figure supplement 1A*). However, cluster 3 uniquely had increased expression of pro-inflammatory markers. Clusters 5 and 6 harbored gene profiles of multipotent stem and progenitor cells (*Dpp4, Cd34, Ly6a* (encoding Sca-1), and *Pi16*) and did not express differentiated adipocyte markers such as *Fabp4*; most of these markers displayed higher expression in cluster 5 than

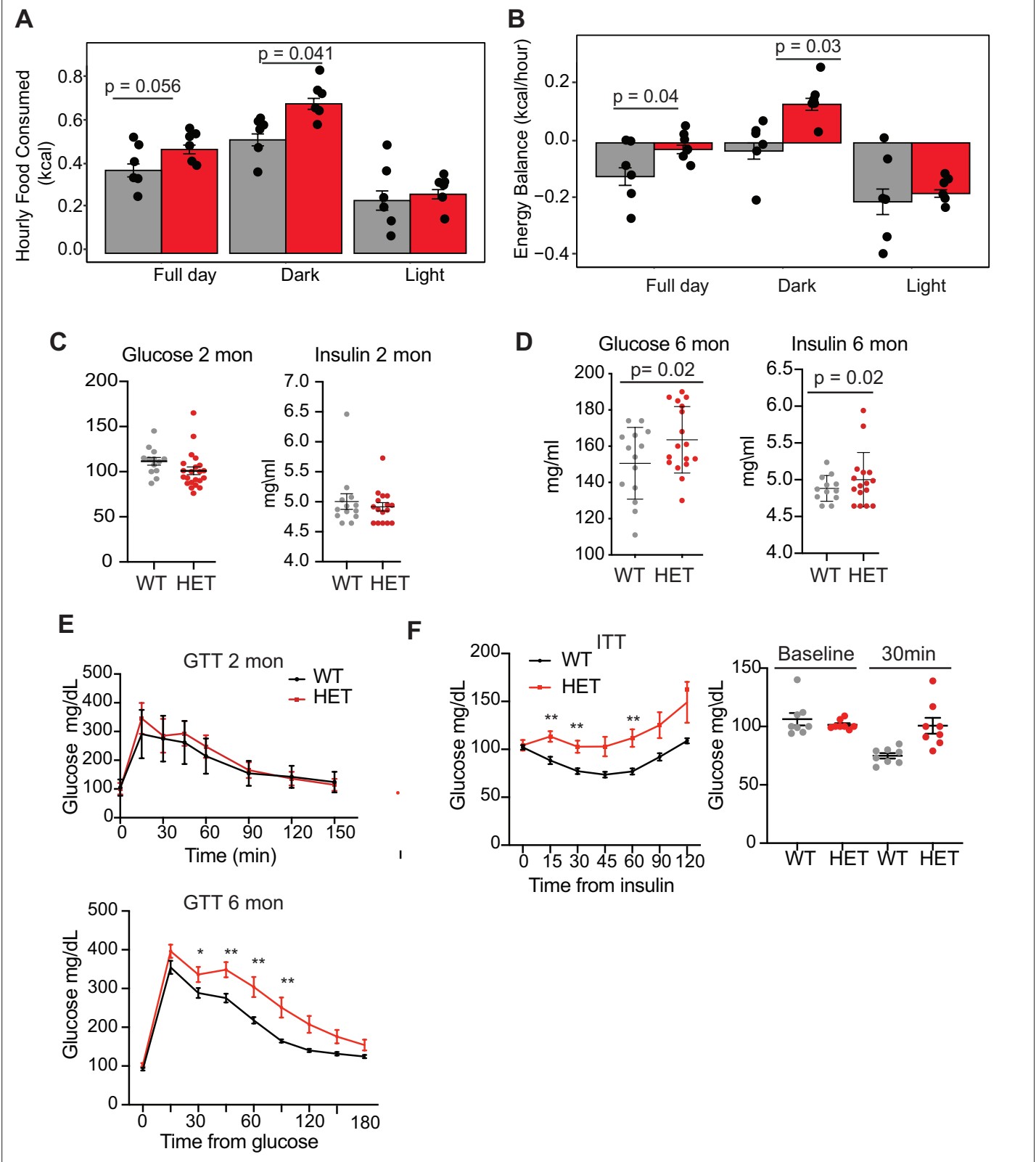

**Figure 2.** *Dnmt3a*-HET mice display increased feeding and develop glucose and insulin intolerance. (**A**) Average hourly food consumption for WT (3 males and 3 females) and HET mice (3 males and 3 females) as averaged for data collected over 5 days. All analyses for CLAMS was performed using CalR and the Statical analysis ACNOVA. (**B**) Energy balance calculated from calories consumed versus calories expended. For the mice as in a. (**C**) Levels of fasting glucose and insulin of WT and HET mice at 2 months of age. All data are expressed as mean ± SEM. (**D**) Levels of fasting glucose and insulin

*Figure 2 continued on next page*

*Figure 2 continued*

of WT and HET mice at 6 months of age. All data are expressed as mean ± SEM. (**E**) Top: Glucose tolerance test at 2 months of age (n=10). Mice were fasted for 12 hr prior to beginning the test. Bottom: Glucose tolerance test at 6 months of age (n=10). Mice were fasted for 12 hr prior to beginning the test. All data are expressed as mean ± SEM. *p<0.05, **p<0.01, ***p<0.0001 by two-way ANOVA. (**F**) Left: Insulin tolerance test at 10 months of age (n=10). Mice were fasted for 12 hr prior to beginning the test. Right: scatter plot of glucose levels at baseline and at 30 min for the mice used in the analysis. All data are expressed as mean ± SEM. *p<0.05, **p<0.01, ***p<0.001 by two-way ANOVA.

The online version of this article includes the following figure supplement(s) for figure 2:

**Figure supplement 1.** Dnmt3a-HET mice display altered metabolic activity.

cluster 6 (*Figure 3B, C*, *Figure 3—figure supplement 1A*). With these data and the gradient in *Dpp4* and *Fabp4* expression across clusters (*Figure 3E*), we classified three major groups of cells: committed preadipocytes (COM; clusters 1–2, with high *Fabp4*), primed preadipocytes PRI; (clusters 3–4, with both stem-like and differentiation markers), and stem and progenitor clusters (ST; clusters 5–6).

We next examined how the cellular composition of WT and 3A-HET adipose tissue changed over time. We calculated the proportion of WT and 3A-HET cells in each of these cellular states at 8 weeks and after 1 year. We found that at young ages, the proportions of stem and primed preadipocytes progenitors were higher in 3A-HET mice (1.35- and 1.5-fold, respectively) compared to WT, while the proportion of cells in the committed clusters was lower than in WT (1.4- fold) (*Figure 3F*, *Figure 3—figure supplement 1E*). These differences in the stem and progenitor-primed clusters indicate that young 3A-HET mice harbor a larger pool of stem and progenitor cells at the expense of committed adipocytes.

After 1 year, when 3A-HET mice are obese, cells in committed clusters decreased proportionally in WT but increased in 3A-HET iWAT (*Figure 3F*). In addition, we detected a decrease in the ST stem cluster in 1-year-old 3A-HET compared to WT mice, suggesting that adipose expansion in HET mice ultimately results in a decrease in the stem and progenitor clusters, possibly due to exhaustion or increased proliferation.

To validate these observations in cellular proportions, we examined changes with age in immuno-phenotypically defined progenitors. Consistent with our scRNAseq data, we found that, though preadipocyte cells were more abundant in 3A-HET mice at 8 weeks relative to WT, by 1 year their numbers decreased relative to WT (*Figure 3G*). These results imply that intermediate stem and primed adipocytes are the primary source for adipose tissue expansion in WT mice, while in 3A-HET mice, increased energy intake and hyperphagia continuously require more preadipocyte progenitors, ultimately resulting in depletion of stem cells.

## Loss of DNMT3A leads to an inflammatory phenotype in adipose tissue

To better understand which cell types differed in 3A-HET mice and how they changed over time, we used pseudotime analysis of the scRNA-seq data to infer relationships among the cell type clusters. This analysis revealed two major trajectories (A and B), with cluster 5 representing the most naïve position (*Figure 4A, B*). Trajectory A included stem-like cluster 6, and primed (cluster 4) and committed preadipocytes (cluster 2, *Fabp4*^high). Trajectory B passed through clusters that displayed hallmarks of the inflammatory response including upregulated interleukin 6 (*Il6*) and *Ccl2* (C-C motif chemokine ligand 2) (*Figure 3B* and 6C, *Figure 4—figure supplement 1*). Trajectory B also included the pro-inflammatory primed (cluster 3) ending in committed preadipocytes (cluster 1).

Next, we calculated the proportion of WT and 3A-HET cells residing along each of the trajectories. Young WT cells were weighted toward trajectory A, while young 3A-HET cells were much more heavily represented along the pro-inflammatory B trajectory, with more cells retaining stem-like and primed preadipocyte phenotypes (*Figure 4D*). Importantly, at 8 weeks of age, WT and 3A-HET mice did not display overt differences in adiposity, indicating these cell-state differences precede the phenotypic alterations.

Examining the trajectories after 1 year, we noted in WT mice more cells in the pro-inflammatory path, consistent with the known inflammatory effects of aging in adipose tissue (*Tchkonia et al., 2010*). Strikingly, the 3A-HET preadipocytes showed a particularly strong bias toward pro-inflammatory trajectory B (*Figure 4D*). Altogether, these analyses suggest that cluster 5 serves a source for both pathways, and 3A-HET adipose tissue, even at a young age, displays a bias towards an inflammatory state, which becomes exacerbated in aging obese preadipocytes.

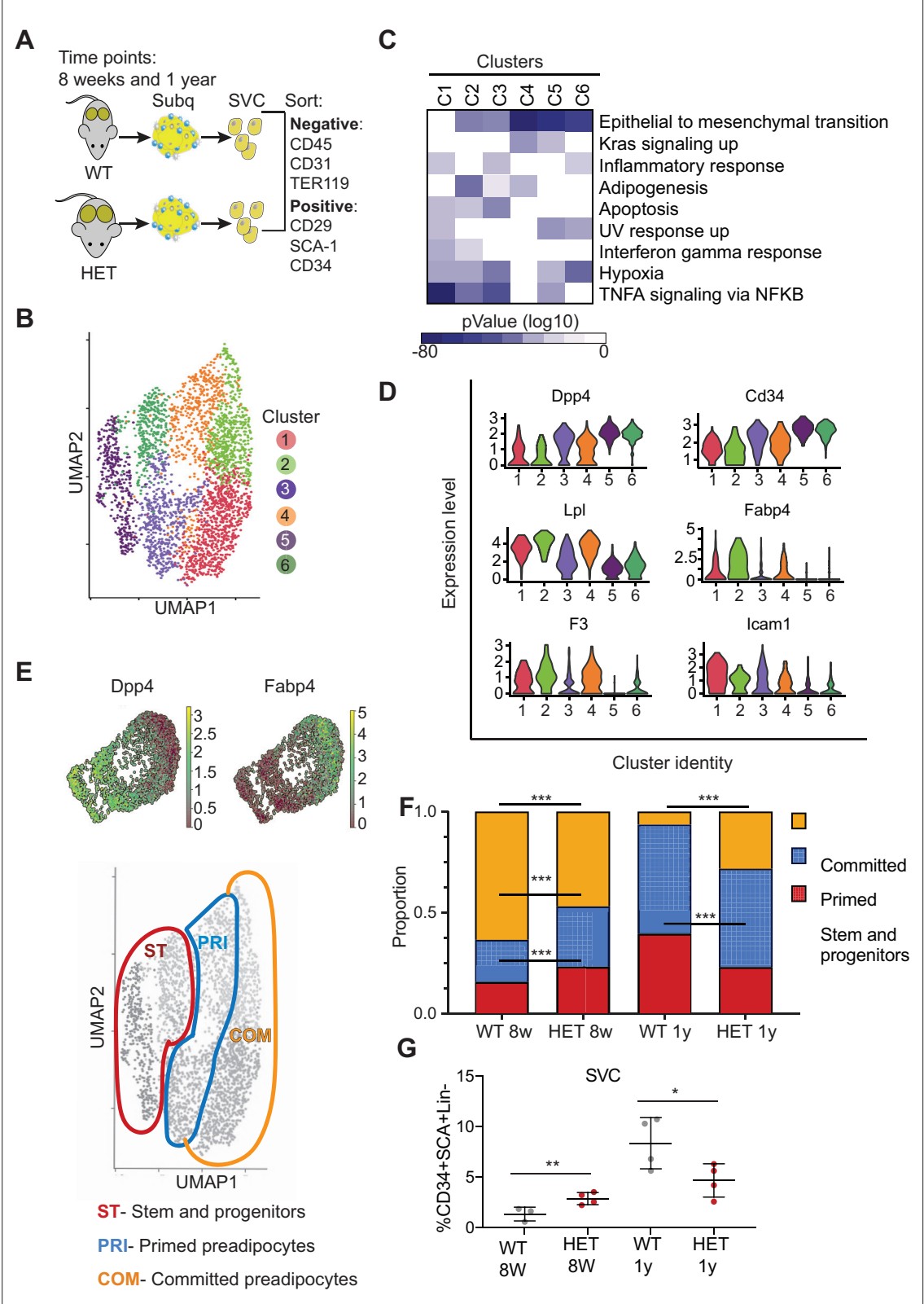

**Figure 3.** *Dnmt3a*-HET adipose tissue contains more stem cell-like preadipocytes at the expense of committed preadipocytes. (**A**) Scheme of single cell RNA sequencing from stromal vascular cells (SVC) of WT and 3A-HET adipose tissue. (**B**) Uniform Manifold Approximation and projection (UMAP) clustering of pooled preadipocyte cells from WT and 3A-HET at 8 weeks and at 1 year of age. (**C**) Top hallmark pathways of marker genes for each cluster shown as a heat map. Marker genes were enriched in a cluster compared to all other clusters. (**D**) Expression levels of cluster-defining marker

*Figure 3 continued on next page*

*Figure 3 continued*

genes depicted by violin plots. The y-axis marks the normalized log-scale of average read counts. (**E**) Top: UMAP representation of the stem cell marker *Dpp4* and the committed preadipocyte marker *Fabp4* across all clusters. Bottom: definition of cellular states for the clusters based on the levels of *Dpp4* and *Fabp4*. Stem and progenitor cells had the highest *Dpp4* with no *Fabp4*. Primed preadipocytes expressed both *Dpp4* and *Fabp4*, and committed preadipocytes expressed mainly *Fabp4*. (**F**) Quantification of the denoted cluster proportions at 8 weeks (8 w) and 1 year (1y). *p<0.05, **p<0.01, ***p<0.001 by Fisher exact test. (**G**) Quantification of preadipocytes in stromal vascular cells (SVC) cells by flow cytometry using the following strategy: cells were selected as lineage negative (lacking Cd45, Cd31, and Ter119) and positively expressing Cd34, Sca1, and Cd29 cell in WT and 3A-HET stromal vascular cells (SVCs) at 8 weeks and at 1 year of age. *p<0.05, **p<0.01 by Student's- t- test.

The online version of this article includes the following figure supplement(s) for figure 3:

**Figure supplement 1.** Isolation of cells used for single cell RNAseq analysis.

Because the inflammatory state of adipose tissue correlates strongly with insulin resistance and other co-morbidities of obesity, we examined the inflammatory markers and trajectories more closely. Differential gene expression analysis supported our observation that 3A-HET cells were primed to express higher pro-inflammatory marker genes in the transient and committed preadipocytes groups (*Figure 4E*, *Figure 4—figure supplement 1B, C*). Moreover, 3A-HET preadipocytes displayed activation of inflammatory signaling in the IL6 and interferon gamma signaling pathways even in the stem-like cells (*Figure 6—figure supplement 1B*). To verify that IL6 was upregulated, we performed intracellular flow cytometry analysis of preadipocytes from WT and 3A-HET mice (6 months of age), confirming markedly elevated IL6 (*Figure 4F*).

Along with the 3A-HET inflammatory phenotype, genes representing full differentiation of adipocytes (*Apoe, Lpl, Fabp4, and Igfbp3*) were expressed at lower levels in 3A-HET cells in the primed and committed cell clusters relative to that of WT tissue at both time points (*Figure 4—figure supplement 1d*). In addition, unlike WT cells which displayed an epithelial to mesenchymal transition (EMT) signature only in the stem cell clusters, 3A-HET tissue displayed enrichment of stem-like EMT even in the primed and committed clusters, suggesting a failure to suppress the stem cell program in 3A-HET preadipocytes (*Figure 4E*). Together, these results support a role for DNMT3A in adipocyte precursor cell differentiation and maturation. Our data also indicate that even heterozygous loss of *Dnmt3a* in preadipocytes promotes pro-inflammatory obesity.

## DNMT3A promotes differentiation of preadipocyte progenitor cell lines

Our single-cell analysis showed that differentiating 3A-HET preadipocytes upregulate inflammatory pathways, while improperly activating adipogenesis. However, concomitant absence of DNMT3A in the hypothalamus and the adipose tissue in germline *Dnmt3a* KO mice confounds interpretation of a specific role for DNMT3A in adipocyte progenitors. Therefore, here we sought to identify if loss of DNMT3A resulted in similar transcriptional and phenotypic consensuses in WAT and BAT preadipocyte cell lines. Therefore, we generated *Dnmt3a*-knockout (KO) clonal cell lines in established white (3T3-L1) and brown (BAC-C4) preadipocyte models (*Figure 5A* and *Figure 5—figure supplement 1A*, *Figure 5—source data 3*).

First, we compared growth rates of 3T3-L1-control and 3T3-L1-*Dnmt3a*-KO (3T3-Con and 3T3-KO, hereafter) and of BAC-C4 control and BAC-C4-*Dnmt3a*-KO (BAC-Con and BAC-KO). In both lineages, *Dnmt3a*-KO clones exhibited higher proliferation rates than controls (*Figure 5B* and *Figure 5—figure supplement 1B*), mirroring the expansion of 3A-HET progenitor cells in our scRNAseq analysis. In response to hormonal stimulation, 3T3-L1 and BAC-C4 terminally differentiate into adipocytes that accumulate triglycerides in the form of lipid droplets. To analyze the role of DNMT3A in adipocyte differentiation, we quantified lipid droplet size and number in our control and knockout lines. We found that all four lines differentiated and formed mature adipocytes that stored lipids, but both 3T3-KO and BAC-KO stored fewer lipids (*Figure 5C* and *Figure 5—figure supplement 1C*). In addition, both KO cell lines accumulated fewer lipid droplets when challenged with exogenous fluorescent fatty acids, indicating that loss of *DNMT3A* impairs lipogenesis (*Figure 5D* and *Figure 5—figure supplement 1D*).

In addition to lipogenesis, adipocytes breakdown lipids through lipolysis (*Ertunc and Hotamisligil, 2016*). To test the role of DNMT3A in lipolysis, we treated WT and KO cell lines with isoproterenol, a non-selective β-adrenergic receptor agonist. We observed slowed release of fatty acids in both KO lineages with no effect on cell numbers (*Figure 5E*, *Figure 5—figure supplement 1E*). Moreover,

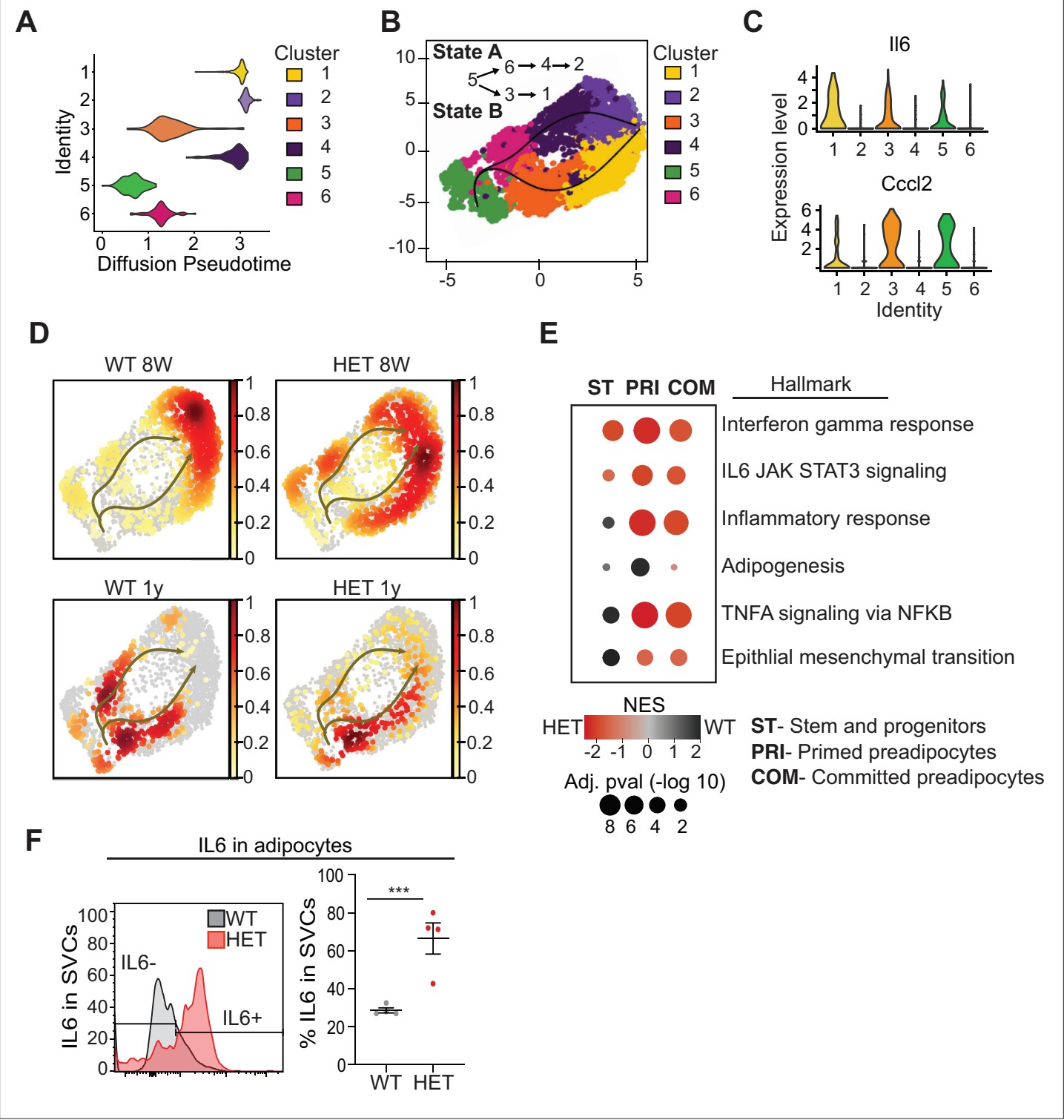

**Figure 4.** Pseudotime diffusion analysis shows that loss of Dnmt3a leads to increase in pro-inflammatory cellular trajectory. (**A**) For each of the clusters as identified in 8-week-old WT mice preadipocytes we calculated diffusion score. The lowest diffusion value indicates that cells within the cluster display a more naïve stem-like gene expression profile. (**B**) Cell trajectories inferred by slingshot analysis overlaid on UMAP data (curved lines). (**C**) Expression levels of inflammatory marker genes that define clusters 1, 3 and 5, as depicted by violin plots. The y-axis marks the normalized log-scale of average read counts. (**D**) UMAP representation of the identified cell sates from B for cells from 8-week and 1-year-old mice. (**E**) Gene enrichment analysis (MsigDB) based on differential gene expression as calculated from the single cell RNA seq data. For each of the specified groups we ranked genes by average log2 fold change within state A or B and separated by genotype. Color indicates relative contribution of WT (blue) or HET (red) cells. Size of

*Figure 4 continued on next page*

*Figure 4 continued*

circle denotes bonferroni log transformed 10 adjusted p-value. (**F**) IL6 protein levels in permeabilized SVCs that lack CD45 marker by flow cytometry. Left: representative quantification of Il6-positive cells. Bottom: quantification of Il6 positive cells as percentage of CD45-negative cells in adipocytes from 1-year-old mice.

The online version of this article includes the following figure supplement(s) for figure 4:

**Figure supplement 1.** Identification of cell cluster marker genes.

phosphorylation of the key lipase regulator hormone sensitive lipase (HSL) was significantly reduced in both KO lineages (*Figure 5F*, *Figure 5—figure supplement 1F*), consistent with reduced lipolysis. Combined, these results indicate loss of *DNMT3A* in preadipocytes allows at least partial differentiation but impairs both lipogenesis and lipolysis responses in WAT and BAT.

To broadly examine the DNMT3A-regulated differentiation program in BAT, we performed RNAseq on BAC-Con and BAC-KO cells and identified differentially expressed genes in non-treated (NT), differentiated, and isoproterenol treated cells (ISO). We identified ~1000 genes that were significantly differentially expressed between BAC-Con and BAC-KO cells following differentiation (*Figure 5—figure supplement 1G* and *Figure 5—source data 1*). When we classified genes upregulated in KO compared to control following differentiation or ISO treatment, we noticed that 30% (244 genes, *Figure 5—figure supplement 1G* left) were already differentially expressed in NT cells (*Figure 3G* top). In contrast, only a small number (41 genes, *Figure 5—figure supplement 1G* right) of genes that were upregulated in control cells following differentiation were already altered at baseline in KO cells.

When we plotted the average expression of genes differentially expressed in control or KO cells following 5 days differentiation, these genes were broadly altered in the same direction in the untreated KO cells (*Figure 5G, H*). The large number of genes not downregulated in KO cells during differentiation indicates that DNMT3A is needed for gene repression during adipocyte differentiation. Pathway enrichment analysis showed that while genes upregulated in control cells during differentiation were enriched for adipogenesis, KO cells displayed enrichment for EMT and inflammation (*Figure 5G, H*).

To determine if loss of *DNMT3A* similarly effected gene expression in WAT, we examined data from differentiating 3T3-L1 cells following *Dnmt3a* knock-down (*You et al., 2017*). Genes upregulated in BAC-KO significantly overlapped with those in 3T3-L1 siRNA-*Dnmt3a* (163 genes, *Figure 5—figure supplement 1H* top). Pathway enrichment of overlapping genes differentially expressed due to *Dnmt3a* loss in BAC also indicated congruence with in the 3T3-L1 cell line, including upregulation of EMT and inflammation (*Figure 5—figure supplement 1H* bottom; *Figure 5—source data 2*). The EMT gene signature is consistent with a mesenchymal stem cell state, indicating that loss of *DNMT3A* causes both BAT and WAT to retain a preadipocyte-like gene expression pattern despite at least partial induction of differentiation. These data establish a cell-autonomous role for DNMT3A in maturation of both white and brown adipocytes.

## Loss of DNMT3A perturbs similar transcriptional pathways in vivo and in vitro

To gain further support for an in vivo role for DNMT3A in adipocyte maturation, we sought to compare the changes observed in WAT and BAT cell lines lacking *Dnmt3a* with changes observed in our single-cell data from adipocyte progenitors. We performed GSEA analysis and overlapped the normalized enrichment score (NES) from both datasets (*Figure 6—source data 1* and *Figure 6—source data 2*). We compared the undifferentiated BAC progenitors to the in vivo population of cells we identified as 'primed', and the differentiated BAC progenitors to the 'committed' populations as identified for WT and HET in the scRNAseq (*Figure 3E*). Our results show that key gene hallmarks differentially expressed in the preadipocyte cell lines (BAC and 3T3) include gene sets identified in our single cell data, such as inflammatory, IL6 and epithelial mesenchymal transition (*Figure 6A, B*). Expression of these gene pathways is consistent with incomplete differentiation and upregulation of inflammation following loss of DNMT3A both in vitro and in vivo.

Our cell line data indicate that DNMT3A is required for induction of a functional lipolysis program. Regulation of lipolysis is important to maintain energy levels during fasting. To determine whether impaired lipolysis impacted free fatty acid (FFA) release in serum, we fasted 10-month-old WT and 3A-HET mice for 12 hr. We detected less FFA and glycerol in the plasma in 3A-HET mice compared to

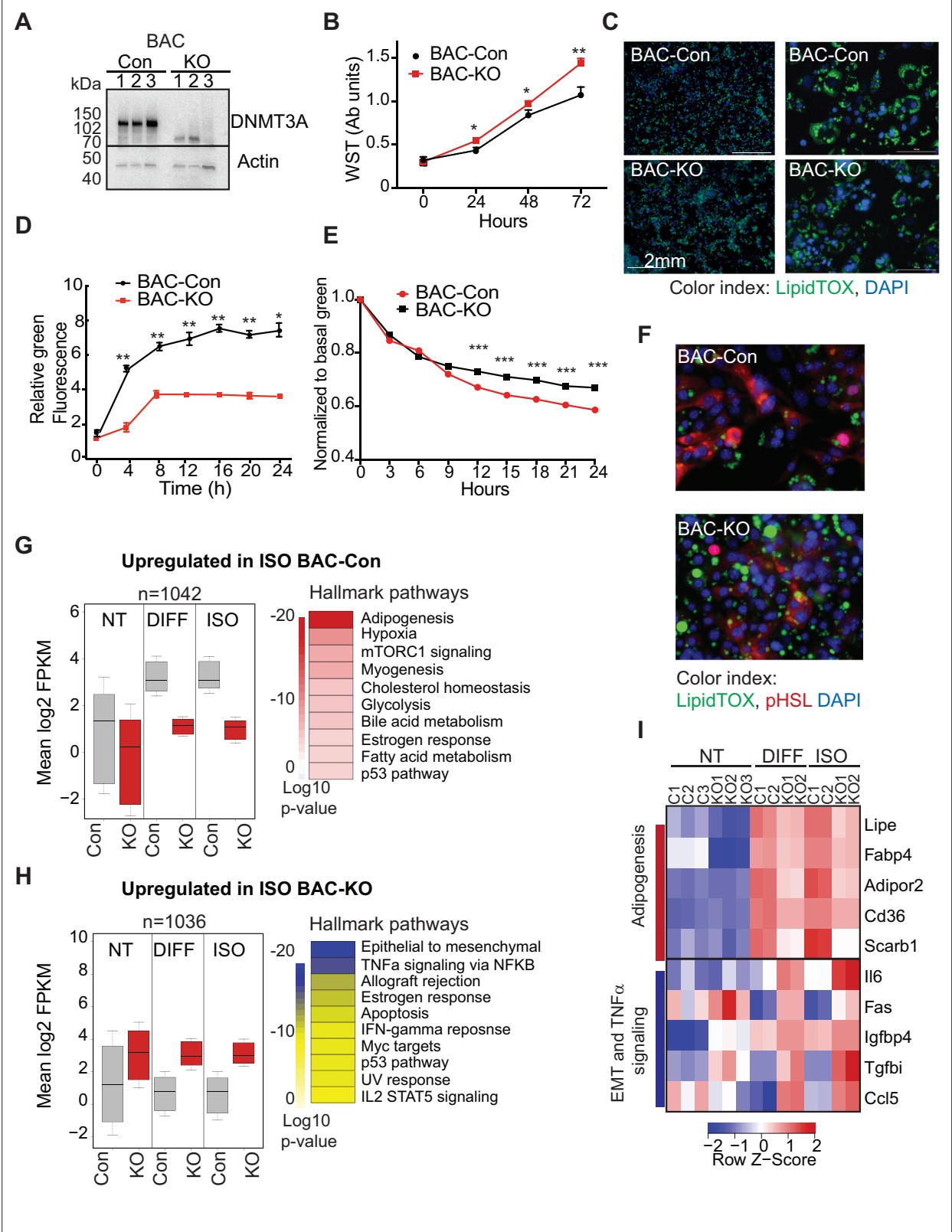

**Figure 5.** DNMT3A regulates preadipocyte proliferation and differentiation. (**A**) Western blot probed with DNMT3A antibody after CRISPR KO in BAC-C4. Three clones of each are shown. (**B**) Proliferation of three WT and KO clones of BAC-C4 cells; assay based on water-soluble tetrazolium (MTT). (**C**) Representative staining of lipid droplets (LipidTOX (green)) following 6 days of differentiation of BAC-C4 control and KO cells. Scale 2 mm. (**D**) Incorporation of fluorescent fatty acid (BODIPY FL C$_{12}$ (green)) relative to red nuclear fluorescent staining quantified during live cell imaging at

*Figure 5 continued on next page*

*Figure 5 continued*

day 5 of differentiation. Statistical analysis two-way ANOVA. *** p<0.0001 (**E**) Quantification of green, fluorescent fatty acid in control and KO cells following treatment with isoproterenol. Data shown is relative to the abundance of green fluorescence at baseline. Statistical analysis two-way ANOVA. *** p<0.0001 (**F**) Representative immunostaining of cells with the indicated antibodies. Cells were differentiated for 5 days and then treated with isoproterenol for 24 hr (**G**) RNAseq analysis at baseline (average of three clones from each genotype, non-treated, NT), at day 6 of differentiation (DIFF, two clones each genotype), and following 24 hr treatment of cells after 5 days of differentiation with isoproterenol (ISO, two clones each). Left panel: For genes defined as upregulated in BAC-control compared to KO following ISO, we plotted log transformed mean expression for all conditions. Right panel: Heatmap of MsigDB enrichment of genes upregulated in BAC-control (fold >2, p ≤ 0.05) compared to BAC-KO (Fold >2, p ≤ 0.005). (**H**) Using RNAseq data from (**E**): Left panel: log transformed mean expression for all conditions for genes defined as upregulated in BAC- KO compared to control following ISO. Right panel: Heatmap of MsigDB enrichment of genes upregulated in BAC-KO (fold >2, p ≤ 0.05) compared to BAC-Control (Fold >2, p ≤ 0.005). (**I**) Heatmap of top 5 genes differentially expressed between control and KO and annotated to the hallmarks adipogenesis or Epithelial to mesenchymal transition (EMT) and TNFa signaling pathways.

The online version of this article includes the following source data and figure supplement(s) for figure 5:

**Source data 1.** RNAseq data from control and Dnmt3a-KO BAC-C4.

**Source data 2.** List of differentially expressed genes overlapping with BAC-C4 and 3T3L1.

**Source data 3.** Western blot probed with DNMT3A antibody after CRISPR KO in BAC-C4 cells. Three clones each of WT and DNMT3A-KO cells are shown.

**Figure supplement 1.** DNMT3A is needed for proper lipogenesis and lipolysis.

**Figure supplement 1—source data 1.** Western Blot of DNMT3A; source data.

WT (~2 fold; *Figure 6C, D*). This also correlated with decreased pHSL (*Figure 6—figure supplement 1*). To address the role of DNMT3A directly in lipolysis, we also stimulated lipolysis ex vivo with ISO treatment of adipose fat pads extracted from 6-month-old WT and HET mice and measured release of FFA and glycerol. 3A-HET tissues displayed lower secretion of FFA from both BAT and WAT (gonadal, gWAT) compared to WT (*Figure 6E*, *Figure 6—figure supplement 1*). In addition, 3A-HET WAT and BAT tissues released less glycerol compared to WT (*Figure 6F*, *Figure 6—figure supplement 1*). Consistent with our in vitro data, western blot analysis of 3A-HET fat depots showed diminished activation of p-HSL compared to WT in response to ISO or fasting (*Figure 6—figure supplement 1D, E*). Together, these findings demonstrate that, similar to homozygous loss of *Dnmt3a* in cell lines, heterozygous loss of *Dnmt3a* in vivo reduces lipolytic capacity and subsequent FFA release.

## Ablation of *Dnmt3a* in subcutaneous preadipocytes leads to adipose tissue expansion

Our results thus far suggest a direct role for DNMT3A in regulation of adipose tissue formation. To further test for a cell autonomous role for DNMT3A in adipocyte development, we conditionally ablated *Dnmt3a* using transgenic mice expressing Cre under the control of the *Prx* enhancer (*Logan et al., 2002*). *Prx* is expressed in the uncommitted mesenchymal progenitors of subcutaneous inguinal fat, which were the focus of the analyses described above (*Eguchi et al., 2011*; *Logan et al., 2002*).

We monitored mice that were either control (Dnmt3a^flox/flox) or homozygous *Dnmt3a* knockout (PRX-D3A) in inguinal fat for 8 months. We observed a steady yet mild increase in weight (~14%), which became statistically significant at 8 months in PRX-D3A mice (*Figure 7A* and *Figure 7—figure supplement 1A, B*). Body composition analysis confirmed that PRX-D3A mice displayed elevated fat mass compared to controls (*Figure 7B*). In addition, like HET mice, PRX-D3A mice also displayed larger adipocytes in the subcutaneous inguinal fat in which *Dnmt3a* is ablated, while maintaining similar adipocyte size in gonadal fat in which the *Prx-Cre* is not expressed (*Figure 7C*). Consistent with expansion of adipose tissue in these mice, we also detected an increase in plasma leptin in PRX-D3A mice (*Figure 7D*). These results support our hypothesis that loss of DNMT3A specifically from adipocyte progenitors contributes to expansion of adipose tissue mass.

*Prx*-Cre is expressed in inguinal fat and not visceral fat (*Krueger et al., 2014*). Thus, if DNMT3A acts autonomously in adipocyte progenitors, we expected to observe expansion of the adipocyte progenitor pool in inguinal but not visceral fat depots. We performed flow cytometry analysis of SVCs extracted from fat pads of subcutaneous inguinal and visceral fat and indeed observed a significant increase in the percentage of Sca1 +preadipocytes progenitors only from the subcutaneous inguinal fat, supporting cell autonomous expansion of adipose progenitors with loss of DNMT3A (*Figure 7E*).

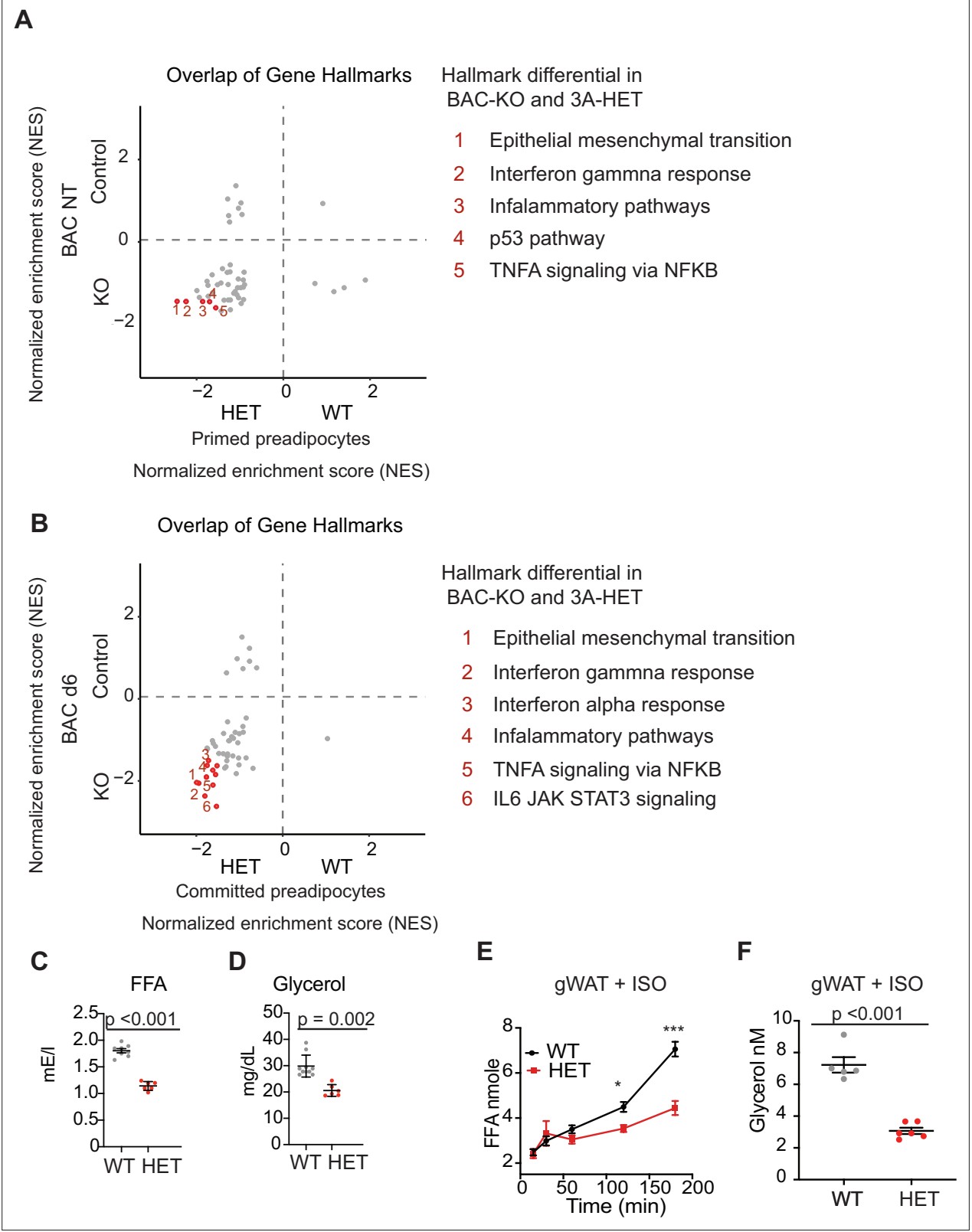

**Figure 6.** DNMT3A regulates lipolysis and inflammatory gene pathways. (**A**) Overlap of normalized enrichment score (NES) of GSEA hallmark pathway overlap between the scRNAseq data for the primed preadipocytes and BAC RNAseq at the nontreated (NT) state. For each dataset, we performed GSEA analysis and calculated NES score. Positive values are pathways enriched for WT adipose and BAC control cells. Negative values are pathways enriched for HET adipose and BAC KO cells. (**B**) Overlap of normalized enrichment score (NES) of GSEA hallmark pathway overlap between the

*Figure 6 continued on next page*

*Figure 6 continued*

scRNAseq data for the committed preadipocytes and BAC RNAseq at day 6 of induced differentiation. For each dataset, we performed GSEA analysis and calculated NES score. Positive values are pathways enriched for WT adipose and BAC control cells. Negative values are pathways enriched for HET adipose and BAC KO cells. (C) Quantification of free fatty acid in the plasma of 12 hr fasted mice. (D) Quantification of glycerol in the plasma of 12 hr fasted mice. (E) Quantification of free fatty acid release from gonadal fat (gWAT) from WT and 3A-HE mice fat was isolated from mice fasted for 6 hr and treated with 1 µm isoproterenol for 3 hr. (F) Quantification of glycerol at the end of the experiments described in e.

The online version of this article includes the following source data and figure supplement(s) for figure 6:

**Source data 1.** fGSEA analysis of NT BAC and primed preadipocytes pathway NES.

**Source data 2.** fGSEA analysis of BAC cell following differentiation and committed preadipocytes pathway NES.

**Figure supplement 1.** DNMT3A is needed for in vivo lipolysis.

**Figure supplement 1—source data 1.** Western Blot for pHSL in gonadal white adipose tissue.

**Figure supplement 1—source data 2.** Western Blot for pHSL in brown adipose tissue.

We also measured the efficacy of lipolysis in subcutaneous inguinal fat from PRX-D3A mice. Similarly to our results from D3A-HET mice, we observed a decrease in the release of FAA from the PRX-D3A fat pads (*Figure 7F*). Taken together, these results confirm a role for DNMT3A in regulation of obesity via early stages of adipogenesis.

## DNMT3A-mediated DNA methylation is required for adipose tissue development

To examine how loss of DNMT3A contributes to functional changes in preadipocytes, we examined DNA methylation alterations using whole genome bisulphite sequencing (WGBS) in SVCs at the same time points used for scRNAseq (8 weeks and 1 year). In 3A-HET cells, we observed a mild (2–4%) yet significant ($p < 10^{-15}$) decrease in global methylation compared to WT cells in both time points (*Figure 8—figure supplement 1A*).

As obesity became extreme with age, we sought to compare age-associated changes in both WT and 3A-HET cells. We first determined DNA methylation changes in WT between the two time points, identifying ~1100 regions that were hypomethylated (hypo-DMRA, see methods), and ~2800 regions hypermethylated with age (hyper-DMRA, see methods) (*Figure 8—figure supplement 1B*, *Figure 8—source data 1 and 2*). We then examined the mean methylation distribution of WT hypo-DMRA in 3A-HET cells (*Figure 8A*). Remarkably, these regions tended to be hypomethylated in 3A-HET cells regardless of age (*Figure 8B*). Comparing the mean methylation for these regions between young WT and HET cells showed that ~900 of them (70%) displayed lower methylation values (≥19%, blue peak in *Figure 8C*) in HET cells than WT and most remained hypomethylated at 1 year (*Figure 8D* and *Figure 8—figure supplement 1C*). Therefore, 3A-HET cells lacking one allele of *Dnmt3a* began with lower DNA methylation and failed to experience the dynamic methylation changes typically seen in WT cells with age.

We next examined the role of DNA methylation gains over time, focusing on the WT hyper-DMRA (*Figure 8e*). When we plotted their mean methylation in 3A-HET cells, young cells displayed significantly lower methylation levels than WT, and methylation was generally even lower at 1 year (~1500, 54%). These data suggest that DNMT3A has a major role in methylation of ageing DMRA regions over time (*Figure 8F*). In fact, when we plotted the methylation of the hyper-DMRA in young and old 3A-HET cells, most of them (~1200) remained hypomethylated across time (*Figure 8G*, purple and *Figure 8—figure supplement 1C*). Taken together, these results suggest that DNMT3A is essential to modulate the DNA methylation landscape during adipocyte development and aging, and that even heterozygous loss of *Dnmt3a* blunts these dynamics.

## Loss of DNMT3A induces a pro-inflammatory trajectory in young adipocytes

To identify genes influenced by loss or gain of DNA methylation in WT and 3A-HET cells, we performed gene ontology analysis using Hypergeometric Optimization of Motif EnRichment (HOMER), analyzing hypo-DMRA and hyper-DMRA regions (*Figure 8*). Our analysis revealed that hypo-DMRA were largely associated with inflammatory and metabolic pathways, while hyper-DMRA were enriched for pathways associated with stem cells (*Figure 9—figure supplement 1A, B*).

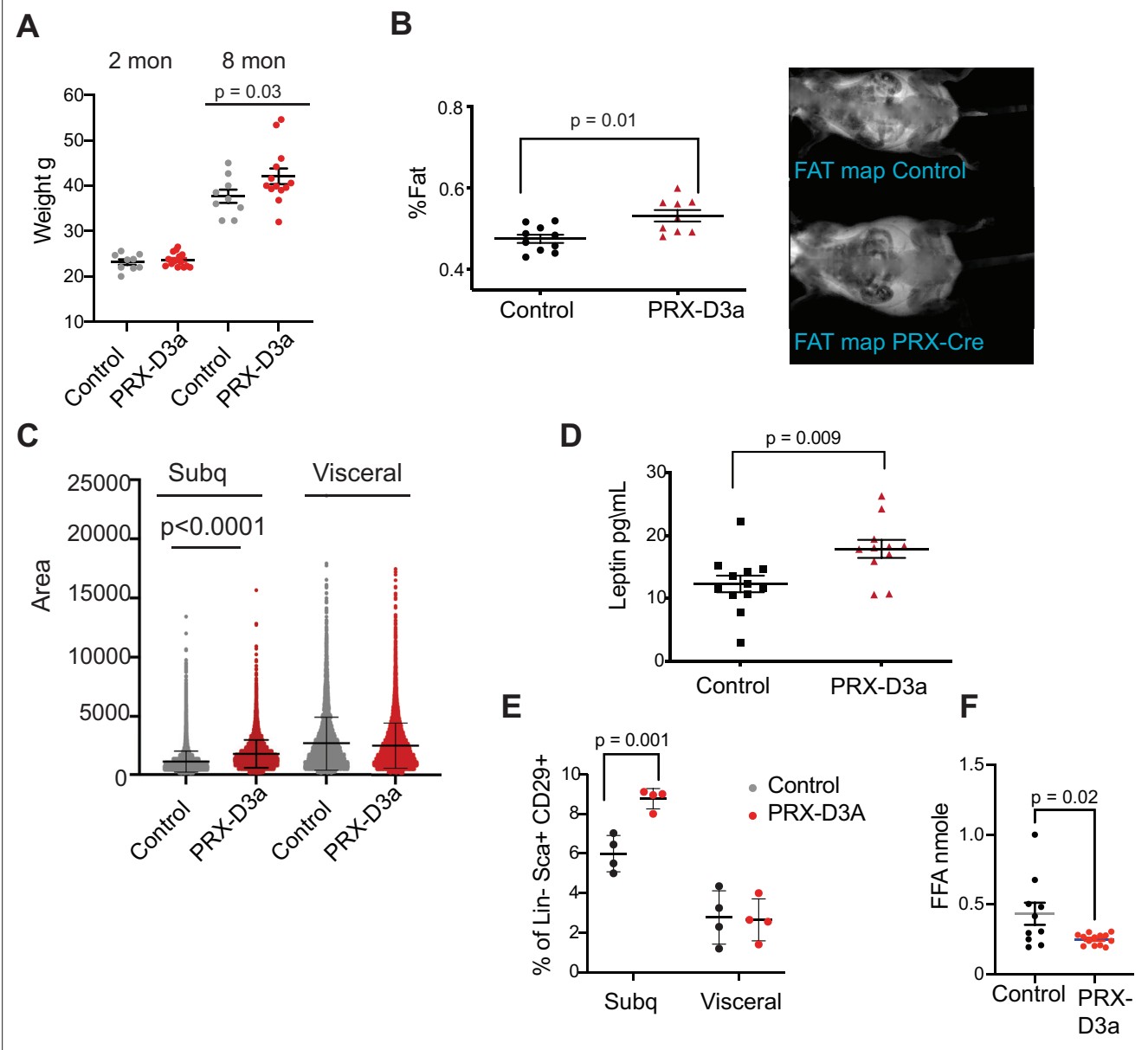

**Figure 7.** Loss of Dnmt3a in inguinal subcutaneous adipose progenitors leads to fat expansion. (**A**) Weight of male *Prx*-Cre *Dnmt3a* mice (PRX-D3A) mice at 8 months of age in compared to controls (Dnmt3a *flox\flox*) (n=8–10). Student T-test was done for statistical analysis. (**B**) Dual energy X-ray absorptiometry (DEXA) analysis of PRX-D3A mice compared to control mice. Left: Quantification of fat mass (n=10–12) from DEXA. Statistical analysis one-way ANOVA. Right: Representative image from DEXA. (**C**) Adipocyte area as measured in histological sections of inguinal WAT in *Prx*-Cre *Dnmt3a* compared to controls. (**D**) Plasma leptin from *Prx*-Cre *Dnmt3a* mice compared to controls. (**E**) Quantification of preadipocytes in stromal vascular cells (SVC) cells from subcutaneous (*Prx*-Cre is expressed) and visceral (*Prx*-Cre not expressed) by flow cytometry using the following strategy: cells were selected as lineage negative (lacking Cd45, Cd31, and Ter119) and positively expressing Cd34, Sca1, and Cd29 cell in *Prx*-Cre *Dnmt3a* mice compared to control. Two-way Anova was done for statistical analysis. (**F**) Quantification of free fatty acids (FAA) release from subcutaneous fat of *Prx*-Cre *Dnmt3a* mice compared to control. Fat pad was isolated from mice fasted for 12 hr and treated with 1 μm isoproterenol for 3 hr.

The online version of this article includes the following figure supplement(s) for figure 7:

**Figure supplement 1.** Ablation of Dnmt3a from subcutaneous fat contributes to weight gain.

In the absence of DNMT3A, lower methylation reflects a failure to establish DNA methylation. Our cell lines and scRNAseq work revealed that 3A-HET cells fail to properly upregulate the adipogenesis gene program while aberrantly increasing a pro-inflammatory cell trajectory. Because WT hypo-DMRA were associated with pro-inflammatory pathways, we hypothesized that lower DNA methylation in

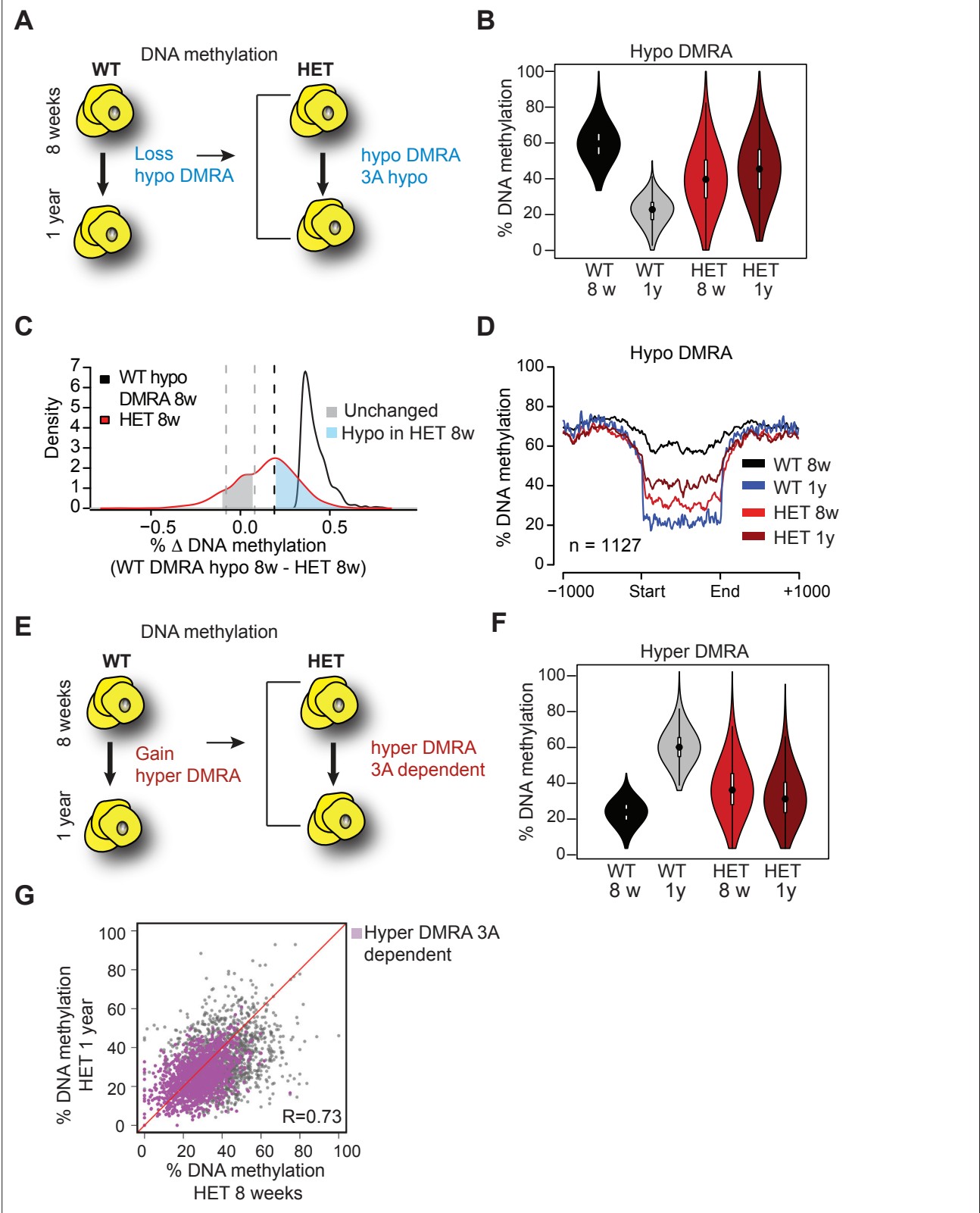

**Figure 8.** Loss of *Dnmt3a* alters DNA methylation landscape during adipocyte development and ageing. (**A**) Schematic for strategy to define regions hypomethylated with age (WT hypo-DMRA). (**B**) Violin plot of mean methylation for WT hypo-DMRA and their mean methylation in 3A-HET. (**C**) For the regions defined as WT hypo-DMRA we subtracted the methylation value for the same loci in 8 weeks HET cells. In this histogram representation, we identified two peaks: in blue are regions that are hypomethylated in HET cells (20% difference or greater) and in grey regions which share similar levels

*Figure 8 continued on next page*

*Figure 8 continued*

in methylation in WT ad HET cells and determined as unchanged regions. (**D**) Mean methylation +/1 1 Kb from the DMR for regions defined as WT DMRA-hypo. (**E**) Schematic of definition of regions hypermethylated in WT ageing (WT hyper-DMRA). (**F**) Violin plot of mean methylation for WT hyper-DMRA in HET cells (8 weeks and 1 year). (**G**) Scatter plot of methylation levels for WT hyper-DMRA and the same loci in HET at 8 weeks and at 1 year. Quantification of the correlation between the regions that remain hypomethylated (20% or larger) in HET cells in 8 weeks and in 1 year (purple) indicates high probability score of them being hypomethylated in HET cells regardless of age.

The online version of this article includes the following source data and figure supplement(s) for figure 8:

**Source data 1.** List of hypo-DMRA.

**Source data 2.** List of hyper-DMRA.

**Figure supplement 1.** DNA methylation in adipocytes across ageing.

those regions in young HET cells would correlate with higher activation of pro-inflammatory trajectory B. We therefore compared gene expression of stem, primed, and committed cell clusters in trajectories A or B separated by genotype to determine in which cell clusters hypo-DMRA led to upregulation of gene expression (average of log2 fold-change; *Figure 9A*). We detected significant correlation for the 3A-HET primed pre-adipocytes, and weaker correlation for committed pre-adipocytes (*Figure 9B*). These results indicate that pro-inflammatory-associated regions that lose methylation during WT aging and are even less methylated in HET cells show increased gene expression in 3A-HET primed and committed cell clusters. Thus, reduced DNA methylation in those regions permits the activation of the pro-inflammatory genes. Examples of genes with this behavior include *Irf1*, *Socs3* and *Jak2* (*Figure 9—figure supplement 1C*). Activation of all these genes in adipocytes previously has been associated with metabolic disease and inflammation (*Friesen et al., 2017*; *Liu et al., 2015*).

We found WT-defined hyper-DMRAs to be regions highly enriched for pathways associated with stem cell and pluripotency, suggesting that the stem cell program is repressed as WT adipose tissue ages and stem cells differentiate (*Figure 5F*). We hypothesized that hyper-DMRAs that did not gain methylation over time in HET cells would be associated with upregulated genes. Indeed, when we calculated the enrichment score of the gene expression profile from our scRNAseq of WT and HET at one year of age, HET cells displayed high correlation with activation of WT hyper-DMRA, which include known stromal progenitor genes such as *F3*, Krueppel-like factor 9 (*Klf9*), protein phosphatase 1 (*Dusp1*) and PDGF receptor β (*Pdgfrb*) (*Figure 9C, D*). Interestingly, *F3* is known for its role in inhibitory adipogenesis regulators (*Schwalie et al., 2018*), suggesting inhibited differentiation in HET cells. To further support a correlation between loss in DNA methylation in HET cells with higher gene expression as found in the scRNAseq, we enriched for adipocyte progenitors in a new cohort of 1-year-old WT and HET mice (n=4). We found increased gene expression for 5 of the genes we measured (*Figure 9—figure supplement 1*). Overall, these results suggest that the failure to gain DNA methylation with age in obese mice promotes enduring expression of stem and progenitor genes, consistent with faulty adipocytes.

## Discussion

Given the constitutional loss of DNMT3A that functions across multiple tissues, our TBRS model offers a unique opportunity to study the association between DNA methylation, adipogenesis, and obesity. Here, we establish a role for DNMT3A in regulating adipogenesis in vivo and in vitro and demonstrate that DNMT3A deficiency predisposes mice to obesity, even on a normal diet. We verified a role for DNMT3A in regulation of feeding and demonstrated for the first time a direct independent role for DNMT3A in regulating adipose tissue formation and function.

We show that heterozygous *Dnmt3a*-null mice recapitulate the obesity phenotype observed in TBRS along with increased length. Our studies show aberrant feeding behavior in HET mice which is in line with previously published data (*Kohno et al., 2014*). In view of the recently established studies on the role of DNMT3A in emotional behavior and anxiety (*LaPlant et al., 2010*; *Morris et al., 2016*), future studies may address the association between DNMT3A loss, complex neuro-psychiatric disorders and feeding behaviors.

Here, we report the particularly striking impact of reduced of DNMT3A on adipose tissue expansion and weight gain. Mutant mice displayed expanded fat depots of all types. Adipocytes were larger with decreased lipolysis as supported by phenotypic and transcriptomic analyses. The trajectory of

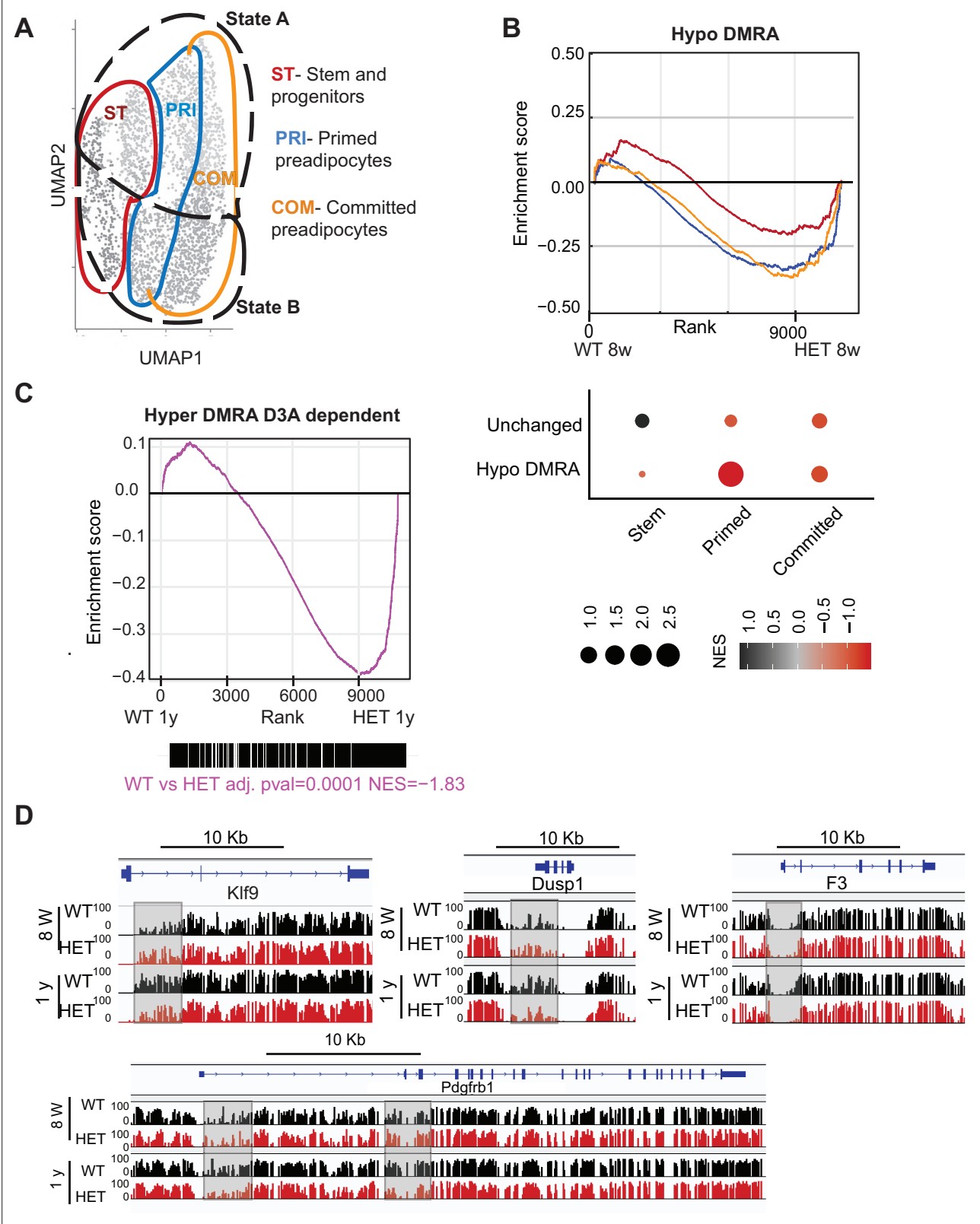

**Figure 9.** Loss of DNA methylation in HET cells is associated with activation of the pro-inflammatory cell state B. (**A**) Illustration of the cell clusters used for the gene enrichment analysis based on the clusters and cellular states identified in *Figure 5*. (**B**) Displayed is the enrichment score and the Bonferroni adjusted p-value for genes associated with WT hypo and HET hypo DMRA based on their enrichment within cell cluster groups and cellular sates as described in A. Bottom: Negative normalized enrichment score (NES) indicates more contribution of HET cells to the genes associated with

*Figure 9 continued on next page*

Figure 9 continued

the WT hypo DMRA regions, while positive NES indicates more contribution of WT cells to the genes associated the described regions in DMRA and in unchanged regions. (C) Displayed is the enrichment score and the Bonferroni adjusted p-value for genes associated with WT hyper and HET (8 weeks and 1 year) hypo DMRA based on their enrichment within cell cluster groups and cellular sates as described in A. Negative normalized enticement score (NES) indicates more contribution of HET cells to the genes associated with the WT hyper DMRA regions, while positive NES indicates more contribution of WT cells to the genes associated the described regions. (D) For regions defined as WT hyper-DMRA and HET 8 weeks and 1 year hypomethylated (purple regions from *Figure 6G*), we performed gene enrichment analysis based on the cell clusters groups and cellular sates illustrated in A. Displayed is the enrichment score and the Bonferroni adjusted p-value in WT and HET cells. (E) DNA methylation at the *Klf9, F3, Dusp1,* and *Pdgfrb* loci in WT (black) and HET (red) cells at 8 weeks and at 1 year. Highlighted are the regions which gain methylation in WT cells but lose methylation in the HET.

The online version of this article includes the following figure supplement(s) for figure 9:

**Figure supplement 1.** DNA methylation alterations in HET cells are associated with adipogenesis and inflammation gene networks.

weight gain is distinct from that of leptin-null mice, which become obese by 4 weeks (*Zhang et al., 1994*). Instead, heterozygous *Dnmt3a* KO mice gain weight gradually, with marked tissue expansion evident mostly after 6 months. Nevertheless, after around 1 year, the HET mice are markedly obese, with body weight typically about a third higher than their WT counterparts. Future studies in which DNMT3A is restored in the HET model may further support the role of DNMT3A in regulation of lipolysis and inflammation in the adipose tissue.

We present evidence here that, in addition to its role in energy homeostasis through the central nervous system, DNMT3A directly regulates fat by enabling proper adipocyte progenitor development. The increased fat after ablation of *Dnmt3a* specifically in inguinal depots supports a cell autonomous function for DNMT3A in preadipocytes. Furthermore, ablation of *Dnmt3a* in white and brown preadipocyte progenitor cell lines led to incomplete differentiation, underscoring the direct role. While knockout of *Dnmt3a* using Adiponectin-Cre, which is specific for fully differentiated adipocytes, did not lead to weight gain (*You et al., 2017*), this is expected given our hypothesis that DNMT3A primarily acts in stem and progenitor cells and not on differentiated cells. This concept was elegantly validated in the hematopoietic system where knockdown of *DNMT3A* in human macrophages had minimal effect compared to its loss in stem cells (*Lim et al., 2021*). These results support the broader concept that a primary role for DNMT3A is in progenitors, enabling proper differentiation, after which it is dispensable.

Our data indicate that loss of DNMT3A expands adipocyte progenitors which exhibit aberrant differentiation: young 3A-HET mice displayed an increase in the preadipocyte pool and fewer committed cells compared to WT, and differentiated cells in vivo and in vitro were less functional by multiple measures. Paradoxically, this led to more adipose tissue and obesity. While counterintuitive, these results are in accord with observations from the hematopoietic system. Loss of DNMT3A leads to a marked increase in stem cell pool. Despite somewhat impaired differentiation, the greater number of stem cells leads to an overall increase in the number of differentiated cells in the peripheral blood (*Challen et al., 2011*; *Jeong et al., 2018*). This imbalance between self-renewal and differentiation is believed to underlie the propensity of DNMT3A mutations to contribute to clonal hematopoiesis and leukemia (*Challen and Goodell, 2020*).

In addition to blunted differentiation, loss of DNMT3A led to a precocious inflammatory phenotype in the adipose tissue. We used scRNAseq and DNA methylation analysis to define features of white adipose tissue over time in WT and HET mice. Similar to previously published scRNAseq studies (*Merrick et al., 2019*), we identified stem-like DPP4[high] cells as a renewable source of adipocytes over time. The adipocyte precursors followed a pro-inflammatory trajectory characterized by elevated IL6 as WT mice aged. Strikingly, this pro-inflammatory program was already activated at 8 weeks in 3A-HET mice, months before development of obesity, suggesting that DNMT3A is required for proper adipocyte differentiation and prevents premature activation of a proinflammatory cascade. Differences in DNA methylation profiles between young and old WT and HET preadipocytes correlated closely with the precocious activation of the inflammatory cascade with reduced DNMT3A.

In obesity, adipose tissue inflammation is triggered by intrinsic signals including hypoxia, adipocyte death, and elevated lipid levels, which in turn exacerbate tissue dysfunction, lipotoxicity, inflammation, and metabolic disease (*Reilly and Saltiel, 2017*). Not only did loss of *Dnmt3a* alter adipocyte development, it also predisposed adipocytes towards a pro-inflammatory state prior to elevation of

lipid levels. These results indicate that reduced DNMT3A impacts preadipocytes similarly to high-fat-diet-mediated obesity. Taken together, our study suggests that DNMT3A is essential for normal adipose tissue function over time, and DNMT3A may safeguard from inflammatory obesity in WT populations.

Given the known role of DNMT3A in regulating energy via the CNS, our data suggest a convergence with its activity in the adipose tissue. The constitutive reduced DNMT3A in TBRS patients creates a domino effect leading to the development of marked inflammatory obesity and metabolic disease. In future studies, it will be important to investigate how other tissues, such as the liver and the hematopoietic system, integrate into this cascade to contribute to obesity and its comorbidities.

## Materials and methods

### Mouse models

All mice were housed in AAALAC-accredited, specific-pathogen-free animal care facilities at Baylor College of Medicine, and all procedures were approved by the BCM Institutional Animal Care and Use Committee under protocol AN2234. Mice of both sexes were used unless stated otherwise, and experimental mice were separated by sex and housed with 4 mice per cage. All mice were immune-competent and healthy prior to the experiments described. Mice were bred and maintained at regular housing temperatures (23 °C) and 12 hr light/12 hr dark cycle starting at 7:00 a.m. Animals had ad libitum access to water and chow diet. For physiological fasting experiments, mice were fasted from 7:00 pm to 7:00 a.m. To generate germline *Dnmt3a* heterozygous haploinsufficient mice we crossed *Dnmt3a^{flox/flox}* mice previously obtained from the Beaudet laboratory (*Challen et al., 2011*) with E2a-cre mice obtained from Jackson labs. To generate adipocyte progenitor-specific depletion of DNMT3A, we also crossed the *Dnmt3a^{flox/flox}* to a *Prx*-Cre obtained from Jackson labs. We focused our analysis on males expressing the *Prx*-Cre in which transgene regulation has high fidelity. Accordingly, all comparisons, with knockout Cre models were done on male mice at 6–8 months of age.

### Cell culture and differentiation of preadipocytes

The murine 3T3-L1 and BAC-C4 preadipocyte cell lines were kindly obtained from the laboratory of Mikhail Kolonnin. The cell lines were validated by testing their ability to differentiate to white or brown adipocytes (respectively) and tested for mycoplasma. For differentiation assays cells were plated at $5 \times 10^5$ until confluent in 10% FBS high glucose DMEM. Following they were induced to differentiate, using the adipose-differentiation cocktail described below. For 3T3L1: cells were treated for 48 hr with induction media (25 mM indomethacin, 1 mM dexamethasone, 0.5 mM IBMX, 1 mM insulin), next media was replaced and supplemented only with 1 mM insulin for an additional 4 days until differentiation was complete and lipid droplets were formed. For BAC-C4: cells were treated for 48 hr with induction media (25 mM indomethacin, 1 mM dexamethasone, 0.5 mM IBMX, 0.5 mM rosiglitazone, 1 mM insulin). Following media was replaced to differentiation media (0.5 mM rosiglitazone 1 mM insulin) for an additional 48 hr, next media was replaced and supplemented only with 1 mM insulin for an additional 48 hr.

### Mouse phenotypic methods

#### Mouse body imaging

For mouse body imaging of fat and lean tissues we used Dual energy x-ray absorptiometry (DEXA) for body composition and Bone Mineral Density at 6 months and 1-year-old mice. To compare changes to bone growth and size we also used CT (3D X-Ray) Trifoil eXplore scanner.

#### Comprehensive assessments of energy balance (CLMAS)

For each CLMAS experiment, we used WT and HET littermates at the age of 20–24 weeks. Both genders were used, 3 for each. Prior to recording of data mice were housed in the CLAMS feeder cages for 24 hr to familiarize themselves with the feeding system and to estimate food intake. Following mice were recorded for 4–5 days. All analyses were done by using CalR and data presented as significant are only parameters that were significant when accounted for body mass (*Mina et al., 2018*).

## Glucose and insulin tolerance test

Mice were fasted for 12 hours, following they received IP injection of 20% glucose per kg, for glucose tolerance test or 0.15 IU/kg insulin for insulin tolerance test. Blood samples are collected prior to and after the injection at time 0, 15, 30, 60, and 120 min via tail vein bleeding. Blood levels of glucose are measured based on using a glucometer. Fasting glucose or insulin levels were measured from plasma by glucose hexokinase assay by using Millipore rat/mouse insulin ELISA kit.

## Histopathology

Animals were scarified, and tissues for downstream analysis were fixed in 4% (vol/vol) buffered formalin and embedded in paraffin. Sections were stained with hematoxylin and eosing using standard protocols. For measurement of adipocytes size and number, we used metamorph on 1 mm sections. Vacuole diameter size of 150 micrometer diameter was set as the upper limit, to avoid collection of broken cells. Quantification of Oil Red O droplets within liver section was done using imageJ.

## Immunoblotting

Cells or minced mouse tissue were lysed with 1 x RIPA buffer supplemented with the Halt Protease and Phosphatase inhibitor cocktail (ThermoFisher) for 30 min. Following, protein concentration was quantified, and lysates were then boiled at 95 C in 1 x Laemmli (Bio-Rad) for 5 min. The proteins were separated by SDS-PAGE on 4–15% gradient gels (Bio-Rad), and transferred onto nitrocellulose membranes (Bio-Rad). After 1 hr of blocking in 3% skim milk, membranes were incubated overnight with the following primary antibodies: anti- actin (Santa Cruz), anti-Dnmt3a (Cell signaling), anti-p-HSL (Cell Signaling), anti-STAT3 (Cell Signaling), anti-p-STAT3 (Cell Signaling). This was followed by secondary antibody incubation with anti-mouse or anti-rabbit horseradish peroxidase-conjugated secondary antibody (Santa Cruz), and imaging on the Bio-Rad ChemiDoc platform. Antibodies used: β-Actin Antibody (Cell Signaling, Cat$4,967 S), phosphor-HSL(Ser660) (Cell Signaling. Cat #4,126 S).

## Live cell staining and immunostaining

For staining of lipid droplets, we used HCS LipidTOX following the manufacturer's instructions (Invitrogen). For live imaging of cells as described for fatty acids up-take we treated cells at day 5 of differentiation with BODIPY (10 μM) $C_{12}$ (4,4-Difluoro-5,7-Dimethyl-4-Bora-3a,4a-Diaza-s-Indacene-3-Dodecanoic Acid) (Invitrogen). In addition, to measure cell growth cells were stained with Incucyte NucLight Rapid Red Reagent for nuclear labeling following the manufacturer's instructions (Sartorius). Those experiments were monitored using Incucyte live cell analysis system. For immunostaining 7 days post differentiation, cells were plated on coverslips for 24 hr and following fixed in 4% paraformaldehyde in PBS for 10 min at room temperature, rinsed with PBS, and permeabilized with 0.2% Triton X−100 in PBS for 5 min at room temperature. Blocking was done with normal goat serum for 1 hr. Following primary antibody phospho-HSL (Ser660) 1:100 (Cell signaling) was added for overnight incubation. All samples were then incubated in secondary antibody 1: 1000, Alexa Fluor 488 conjugate (Abcam), and mounted with ProLong Gold antifade reagent with DAPI (Invitrogen).

## Lipolysis assays

For ex vivo lipolysis, adipose tissue from inguinal, gonadal, or brown adipose tissue were harvested from 6 months old WT and Dnmt3a-HET male mice following 6 hr fasting. Tissues were cut into small pieces of 10 mg in duplicates and placed in Krebs buffer (114 mM NaCl, 4.7 mM KCl, 1.16 mM MgSO4, 1.2 mM KH2PO4, 2.5 mM CaCl2, 5 mM NaHCO3, 20 mM Hepes, 0.2% BSA, 2.5 mM glucose). Next tissues were treated with isoproterenol (10 nM) for 3 hr at 37 °C. Free-fatty-acids (FAA) and glycerol were measured using manufacturer's instructions (Sigma). To normalize for tissue size proteins were quantified for each sample. For measurement of protein content, the tissue was homogenized with RIPA cell lysis buffer (Milipore) and then lysate was quantified with Pierce BCA protein assay kit (Thermo Scientific).

## Stromal vascular cells (SVC) isolation and sort

Adipose tissue from relevant depot harvested were minced in Krebs buffer (described above) supplemented with 1% bovine serum and 0.5 mg/ml collagenase I (Themo Scientific) and dispase (2.4 U/ml) (Sigma) following 45 min incubation at 37 °C. Minced tissues were then washed with Hank's buffered

salt solution (HBSS), supplemented with 10 mM HEPES (GIBCO) and 2% heat-inactivated bovine serum (Corning). Following centrifugation steps (2000 rpm 5 min), the pellet was resuspended in red blood cell lysis buffer (155 mM NH4CL, 12 mM NaHCO3, 0.1 mM EDTA) for 5 min. Following additional washing steps with HBSS, stromal vascular fraction was obtained and used for downstream analysis. For analysis of populations, the following antibodies were used: for removal of lineage positive cells: TER-119-PB (1:100), CD31-PB (1:100), CD45-PeCy7 (1:100). For isolation of preadipocytes, lineage negative cells were then selected for the expression of the following antibodies: Sca1- APC Cy7 (1:100), CD34-FITC (1:100), CD29-PE (1:200), IL6-APC (1:100) All monoclonal antibodies were from BD Biosciences or eBioscience. The LSRII cell analyzer was used for data acquisition, sorting was done using Arial. All data analysis was performed using the FlowJo software.

## Next generation sequencing approach and preparation

All data relevant to this study has been deposited to GEO GSE164892.

## Single-cell RNA sequencing

Sorted SVCs were used to make single cell RNA libraries using the 10 x Genomics Chromium Platform. Libraries were sequence on NovaSeq to obtain minimum of 70,000 reads per cell to achieve ~75% barcode saturation (Illumina). Filtering of samples were done by eliminating cells with mitochondrial contamination (<10%), minimum of 500 genes per cell and minimum of 1000 mRNA molecules. Total number of cells used for final analysis ranged at 3000 total cells. From which:~1100 cells were WT 8 weeks, ~1100 cells Dnmt3a-HET 8 weeks, ~500 cells WT 1 year and ~350 cells Dnmt3a-HET 1 year.

## Single-cell RNA processing

All libraries were analyzed using Seruat pipeline (*Butler et al., 2018*). We integrated samples using scTransform with 3,000 features (parameters: normalization method = sct, dims = 1:20, nfeatures = 3000). Next we regressed cell cycle and mitochondrial content. PCA and Shared nearest neighbors were performed using default Seruat functions with FindNeighbors and with parameters dim = 1:30. For clustering we used the RunUMAP function from Seruat (parament: dim = 1:30, n.epochs=500, min.dist=0.4). Clusters were identified using FindClusters (default). Clusters with low representation across all samples were removed (<10% of population) leading to the removal of two clusters.

## Pseudotime algorithms

To identify trajectories and cell lineages we used the following algorithms: destiny and slingshot. In order to identify the cell cluster to be used at the root of the trajectory analysis we performed destiny (diffusion component analysis). Following we used the DiffusionMap on the principle components as defined on the WT 8 weeks single-cell RNAseq sample (number of PCA 1:20). Slingshot was used next to identify cellular relationships with setting of cluster 5 as the beginning of the node tree (parameters: start = cluster 5, allow breaks = false).

## Single-cell differential gene expression

We used the Seruat function FindAllMarkers (parameters test.use = DESeq2) to identify marker genes unique to each cluster when compared to all other clusters. To identify differential expression between genotypes or conditions, we used FindMarkers (parameters: test.use = DESeq2). All plotting of differential of individual genes were done using either Seruat or scanpy default plotting setup (*Challen et al., 2011*).

## Knockout of Dnmt3a in preadipocytes cell lines

Knockout of Dnmt3a was done using Crispr-CAS9 and dual guides with the previously described ribonucleoprotein (RNP) delivery method (*Gundry et al., 2016*). Guides were synthesized by T7- in vitro transcription according to the manufactures instructions (NEB) (Template for guide 1 and 2 in the table below). For genotyping the following primers were used: 5'- TCTGTGGCATCTCAGGGTGA –3', 5'- CCTCCAATCACCAGGTCGAA- 3'; 5'- AGAGAGTGAGCACAGGCCAT-3'. Following validation of successful deletion, serial dilutions were done to isolate individual cell clones.

## RNA sequencing and analysis

RNA was extracted using RNeasy micro kit (Qiagen) and quantified using Nanodrop. We prepared the Truseq stranded mRNA library using the manufacturer's instructions (Illumina). Libraries were sequenced using Nextseq 500 sequencer. Paired-end RNA-seq reads were mapped to mouse genome (mm10) using Hisat 2. Reads were quantified using either Feature Counts or by calculation of the Fragments Per Kilobase of exon per million fragments mapped (FPKM) values using Cufflinks 2.2.1 as defined per each dataset (*Trapnell et al., 2012*). Genes were defined as significant with FPKM ≥ 10, a fold change ≥ 2 and with student t-test ≤ 0.05.

## Quantitative RNA transcription (qRT) by PCR

Adipose tissue was dissected as described above to obtain SVCs. From SVCs adipocyte progenitors were enriched by magnetic bead depletion of immune (CD45 + cells).

## Whole genome bisulfite sequencing (WGBS) and analysis and quantitative RNA transcription (qRT)

Preadipocytes cells were enriched from white adipose tissue SVCs using magnetic depletion of CD45 and CD31. Following, we extracted DNA and RNA from cell pellets using DNeasy kit and All Prep DNA\RNA micro kit (Qiagen). Isolated RNA was then converted to cDNA and used for qRT-PCR analyses. DNA was fragmented prior to bisulfite conversation using NEBNext fragmentase kit, following the manufacturer's protocol (NEB). We used 10 ng DNA to prepare WGBS libraries with the Swift-whole genome bisulfite kit according to the manufacturer's instruction (Swift). Libraries were sequenced using an IlluminaNextseq 500 sequencer. Trimming of swift libraries was done using Trim Galore. Alignment and bisulfite conversation were done using Bisamrk (*Challen et al., 2011*). The WALT pipeline was used for duplicates removal using duplicate-remover as part of the WALT pipeline. We then used methcounts to calculate the methylation ratio of each CpG covered with at least 5 reads (*Chen et al., 2016a*).

## Analysis and definition of hypomethylated regions (HMRs)

To identify HMRs in the adipose tissue WGBS data we used default settings. Briefly, a two-state hidden Markov model-based method was used for the HMR calling (*Song et al., 2013*). This model compares one state representing HMRs to highly methylated background from the same genomic DNA source. We then calculated the mean methylation of HMRs using the methylation ratio for each CpG (Coverage 5 x per) as calculated above. To identify differentially methylated regions (DMRs) that changed in Dnmt3a-HET cells compared to WT, we pooled all HMRs from both genotypes, and HMRs that were within 200 bp from each other were merged. Within those overlapping HMRs, we defined DMRs that differ between genotypes by an average of 30% for the ageing DMRs and 20% for the DNMT3A dependent DMRs.

## Gene ontology and gene enrichment

We used MsigDB and GSEA for gene enrichment analysis and hallmark pathway enrichment for RNAseq data and other genomic loci (*Chen et al., 2013*; *Subramanian et al., 2005*). For the correlation between DMRA and the scRNA expression data we first identified genes differential (average log2 fold) within groups and cellular states as defined in the results section using the FindMarkers function from Seruat version 3.9.9 (logFC threshold = 0, minimum percentage = 0, bonferroni p-value). Next, we ranked genes by their fold change value and used the fGSEA analysis.

## Differentially methylated regions definition and analysis

Differentially methylated regions were initially defined on bases of the WT adipose methylation profile as either hypomethylated (methylation decreased ≥30%) or hypermethylated (methylation gain ≥30%) with age (hypo-DMRA and hyper-DMRA, respectively). For the purpose of methylation and gene expression correlations the association between the methylation status in those regions in HET adipose were further defined based on *Figure 6C*. For HET adipose we defined: WT hypo-DMRA and HET 8 weeks hypomethylated by at least 19% (blue regions from *Figure 7C*) or regions defined as WT hyper-DMRA and HET 8 weeks and 1 year hypomethylated by at least 19% (purple regions from *Figure 7G*).

## Acknowledgements

We thank C Gillespie and K Turner for critical review. We thank P Saha, N Putluri and CS Ward for their assistance and scientific advice as well as the Baylor College of Medicine Cores: Genomic and RNA Profiling, the Cytometry and Cell Sorting, Metabolomics, and Mouse Metabolism and Phenotyping. We thank the J Tadross, J Warner, B Mahler-Araujo and A Vidal-Puig from the Wellcome Trust-Medical Research Council Institute of Metabolic Science at the UK University of Cambridge for their assistance. This work was supported by NIH grants: HG006348, DK114356, HL130249, RR024574, CA183252, DK092883, AG036695, CA222736, CA125123, CPRIT-RP180672 and from the HHMI James H Gilliam Fellowships. We acknowledge support from the Baylor College of Medicine Advanced Technology Cores, including which are supported in part from the following grants: CA125123, HG006348, HL130249, RP180672.

## Additional information

### Funding

| Funder | Grant reference number | Author |
| --- | --- | --- |
| Howard Hughes Medical Institute | Jaime M Reyes | Jaime M Reyes |
| National Institute of Diabetes & Digestive & Kidney Diseases | DK114356 | Sean M Hartig |
| National Institute of Diabetes & Digestive & Kidney Diseases | DK092883 | Margaret A Goodell |
| National Institute on Aging | AG036695 | Margaret A Goodell |
| National Cancer Institute | CA222736 | Yung-Hsin Huang |
| National Cancer Institute | CA183252 | Margaret A Goodell |

The funders had no role in study design, data collection and interpretation, or the decision to submit the work for publication.

### Author contributions

Ayala Tovy, Conceptualization, Investigation, Project administration, Supervision, Validation, Writing – original draft, Writing – review and editing; Jaime M Reyes, Conceptualization, Data curation, Formal analysis, Investigation, Resources, Software, Visualization, Writing – original draft, Writing – review and editing; Linda Zhang, Investigation, Validation; Yung-Hsin Huang, Carina Rosas, Alexes C Daquinag, Anna Guzman, Raghav Ramabadran, Chun-Wei Chen, Tianpeng Gu, Aaron R Cox, Rachel E Rau, Investigation; Sinjini Gupta, Visualization; Laura Ortinau, Dongsu Park, Investigation, Resources; Sean M Hartig, Conceptualization, Investigation, Writing – review and editing; Mikhail G Kolonin, Conceptualization, Investigation, Resources, Writing – review and editing; Margaret A Goodell, Conceptualization, Funding acquisition, Investigation, Project administration, Supervision, Writing – original draft, Writing – review and editing

### Author ORCIDs

Ayala Tovy http://orcid.org/0000-0002-3621-6642
Jaime M Reyes http://orcid.org/0000-0001-7895-2472
Sean M Hartig http://orcid.org/0000-0002-2695-2072
Margaret A Goodell http://orcid.org/0000-0003-1111-2932

### Ethics

This study was performed in strict accordance with the recommendations in the Guide for the Care and Use of Laboratory Animals of the National Institutes of Health. All of the animals were handled according to approved institutional animal care and use committee (IACUC) protocols of Baylor College of Medicine (protocol AN2234). All mice were housed in AAALAC-accredited,

specific-pathogen-free animal care facilities at Baylor College of Medicine, and every effort was made to minimize suffering.

## Decision letter and Author response
Decision letter https://doi.org/10.7554/eLife.72359.sa1
Author response https://doi.org/10.7554/eLife.72359.sa2

---

## Additional files

### Supplementary files
• Transparent reporting form

### Data availability
All data relevant to this study has been deposited to GEO GSE164892.

The following dataset was generated:

| Author(s) | Year | Dataset title | Dataset URL | Database and Identifier |
|---|---|---|---|---|
| Tovy A | 2021 | Constitutive loss of DNMT3A causes morbid obesity through misregulation of adipogenesis | https://www.ncbi.nlm.nih.gov/geo/query/acc.cgi?acc=GSE164892 | NCBI Gene Expression Omnibus, GSE164892 |

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
