## [Editor Report]

In this manuscript, the authors show that DNMT3A is essential for the regulation of adipocyte lipolysis and organismal metabolism. Focusing on a mouse model with heterozygous null mutations in DNMT3A, as seen in Tatton-Brown-Rahman syndrome in humans, the authors show that animals develop obesity with adipocyte hypertrophy. Using a combination of single cell analysis, additional genetic models, and bisulfite sequencing, the authors propose a mechanism whereby DNMT3A regulates adipocyte differentiation and overall metabolism. This well-written paper will be of broad interest to researchers studying metabolism, development, and epigenetics.

---

## [Decision Letter]

**Decision letter after peer review:**

Thank you for submitting your article "Constitutive loss of DNMT3A causes morbid obesity through misregulation of adipogenesis" for consideration by *eLife*. Your article has been reviewed by 1 peer reviewer, and the evaluation has been overseen by a Reviewing Editor and David James as the Senior Editor. The reviewers have opted to remain anonymous.

Essential revisions:

Addressing the following points would be necessary to support the conclusions made.

1) The authors are not consistent in which adipose depots they study. In Figure 1, they introduce cellular phenotypes in gonadal white fat and brown fat, but then the transcriptomic studies use inguinal white fat. A clearer rationale for these choices needs to be provided. Do 3A-HET mice also have large inguinal white adipocytes? In addition, the cellular models used in Figure 5, don't really add to the manuscript.

2) The in vivo phenotyping is incomplete.

(a) The data from the metabolic cages in Figure 2 (main and supplement) should be shown as a trace over time rather than collapsing into a bar graph. In addition, the data needs to be analyzed using a well validated approach, such as ANCOVA (eg. Mina et al., Cell Met, 2018; Tschop et al., Nat Meth, 2011).

(b) The ITT in Figure 2 does not appear to have worked. Neither group really showed any drop in blood glucose following insulin.

(c) The authors argue that factors beyond altered feeding must explain the increased body weight in these mice. This could be tested definitively with a pair feeding study.

(d) The Prx Cre driven deletion phenotyping in Figure 7 is quite superficial. At a minimum, the authors should perform histology and analysis of cell size as they do for the 3A-HET model.

3) It is an overstatement from the authors to state that heterozygous loss of DNMT3A results in proinflammatory obesity (in reference to data in Figures 3 and 4). If the authors want to make this point, this should be definitely shown either by immunohistochemistry or flow cytometry of adipose tissue and/or by measurement of systemic markers of inflammation.

4) In Figure 6, are the defects in lipolysis rescued by an add-back to restore normal DNMT3A levels?

5) The data from the bisulfite sequencing would be more compelling if the authors did qPCR of a selection of genes that are hypo- or hypermethylated to show that the results from bisulfite sequences are consistent with predicted changes at the mRNA level.

---

## [Author Response]

Essential revisions:Addressing the following points would be necessary to support the conclusions made.1) The authors are not consistent in which adipose depots they study. In Figure 1, they introduce cellular phenotypes in gonadal white fat and brown fat, but then the transcriptomic studies use inguinal white fat. A clearer rationale for these choices needs to be provided. Do 3A-HET mice also have large inguinal white adipocytes?

Initially, most of our analyses were performed on gonadal and brown fat because we observed similar phenotypes across all depots. The transcriptomic analysis was performed on inguinal fat because the Prx-cre (used to ablate Dnmt3a specifically in adipocyte progenitors) is activated specifically in these depots. This approach allowed us to compare our data to that of Merrick et al., (2019) which was the main analysis of adipose tissue at the single-cell level published at the time of our experiment. We have added this rationale to the text and added a new section:

“To examine how loss of Dnmt3a affected these cell populations, we performed single-cell RNA sequencing (scRNAseq) on cells isolated from inguinal white adipose tissue (iWAT) of WT and 3A-HET mice at 8-weeks and 1-year old. We focused on the subcutaneous inguinal adipocytes as this depot was previously investigated by scRNAseq enabling to use published data as reference for cluster annotation (Merrick et al., 2019).”

Nevertheless, to address the reviewer’s concern, we performed an additional analysis on the inguinal white fat. The analysis is consistent with other fat depots: DNMT3A-HET have larger adipocyte area in inguinal fat as well. We have added this measurement in Supplemental Figure 1 f and edited the text accordingly.

“We dissected fat depots (gonadal, and subcutaneous inguinal white adipose tissue and brown adipose tissue) and performed histological analysis to determine size of BAT and WAT adipocytes. On average, 3A-HET mice had heavier fat depots across all measured depots”

In addition, the cellular models used in Figure 5, don't really add to the manuscript.

We kindly disagree with the reviewer as the cellular models in figure 5 help establish a role for DNMT3A in adipocyte biology that is not confounded by effects of Dnmt3a-KO in the brain in the context of the whole animal. In addition, the cell line findings offer important corroboration for our in vivo data, as the gene pathways that are differentially expressed in those preadipocytes cell lines recapitulate our in vivo findings from adipose Cre models.

2) The in vivo phenotyping is incomplete. (a) The data from the metabolic cages in Figure 2 (main and supplement) should be shown as a trace over time rather than collapsing into a bar graph.

We have added the trace over time figures in Supplemental Figure 2 per the reviewer’s suggestion.

In addition, the data needs to be analyzed using a well validated approach, such as ANCOVA (eg. Mina et al., Cell Met, 2018; Tschop et al., Nat Meth, 2011).

We agree with the reviewer’s comment. Our data were in fact analyzed by ANCOVA using the ClaR online tool developed by Mina et al., as suggested by the reviewer. We have made sure to mention this now in the figure legend.

“Average hourly food consumption for WT (3 males and 3 females) and HET mice (3 males and 3 females) as averaged for data collected over 5 days. All analyses for CLMAS was done using CalR and the Statical analysis ACNOVA”

(b) The ITT in Figure 2 does not appear to have worked. Neither group really showed any drop in blood glucose following insulin.

In figure 2F we do see a 30% drop in glucose for WT mice that was not recapitulated in HET mice. We have added a bar graph describing the individual mice to show the drop in glucose levels at the 30 minutes time point post insulin injection (Figure 2f).

(c) The authors argue that factors beyond altered feeding must explain the increased body weight in these mice. This could be tested definitively with a pair feeding study.

We thank the reviewer for this comment. We believe that it is the combination of multiple factors, including the overfeeding, that led to this phenotype as it takes the mice some time to gain more weight compared to their WT littermates. We agree that pair feeding study might be more definitive, but we feel that to observe that effect will take a much longer time, the study will be confounded by other aging-related factors that will be altered by DNMT3A, and this experiment will not likely change our explanation that both feeding as well as adipocyte-intrinsic factors account for the phenotype. We have endeavored to emphasize the view throughout the text that both aspects are relevant to the Dnmt3a-HET obesity phenotype.

(d) The Prx Cre driven deletion phenotyping in Figure 7 is quite superficial. At a minimum, the authors should perform histology and analysis of cell size as they do for the 3A-HET model.

We have sought to improve phenotyping with the suggested experiments. We now show that adipocyte area is significantly increased in the Prx-Cre Dnmt3a KO subq-adipocytes. As controls, we also show that gondal adipocytes, in which the Prx-cre is not expressed, are similar in size.

“In addition, like HET mice PRX-D3A mice also displayed larger adipocytes in the subcutaneous inguinal fat in which Dnmt3a is ablated, while maintaining similar adipocyte size in gonadal fat in which the Prx Cre is not expressed (Figure 7C). Consistent with expansion of adipose tissue in these mice we also detected an increase in plasma leptin in PRX-D3A mice (Figure 7D).”

3) It is an overstatement from the authors to state that heterozygous loss of DNMT3A results in proinflammatory obesity (in reference to data in Figures 3 and 4). If the authors want to make this point, this should be definitely shown either by immunohistochemistry or flow cytometry of adipose tissue and/or by measurement of systemic markers of inflammation.

We show IL6 levels in adipose tissue by flow cytometry and now show increased infiltration of macrophages into WAT in HET mice compared to WT mice (Supplemental Figure 1e). We have also tuned down wording throughout the text and adjusted the title of the inflammatory section to:

“Loss of DNMT3A leads to inflammatory phenotype in adipose tissue”

4) In Figure 6, are the defects in lipolysis rescued by an add-back to restore normal DNMT3A levels?

We thank the reviewer for this comment however for technical reasons we were not able to perform the add back experiments. We have added a comment in the discussion that restoring Dnmt3a function could provide additional support to its role.

“Here we report the particularly striking impact of reduced of DNMT3A on adipose tissue expansion and weight gain. Mutant mice displayed expanded fat depots of all types. Adipocytes were larger with decreased lipolysis as supported by phenotypic and transcriptomic analyses. Future studies in which DNMT3A is restored in the HET model may further support the role of DNMT3A in regulation of lipolysis and inflammation in adipose tissue.”

However, we note that the experiments were done with 2-3 independent Dnmt3a KO clones and results are consistent, so we feel our data support the conclusions. We also performed experiments in vivo to measure lipolysis and observed similar trends in HET mice.

5) The data from the bisulfite sequencing would be more compelling if the authors did qPCR of a selection of genes that are hypo- or hypermethylated to show that the results from bisulfite sequences are consistent with predicted changes at the mRNA level.

Thank you for the suggestion. We performed this experiment in a new group of 1-year old WT and HET mice (n=3-4), and all but 1 gene that were found to be hypomethylated and upregulated in the scRNAseq were also upregulated in the qRT data. We have added these data to Supplemental Figure 9f, and modified the methylation Figure 9 to display representative genes from those validated by qRT.